# DIFFUSION MODELS FOR GAUSSIAN DISTRIBUTIONS: EXACT SOLUTIONS AND WASSERSTEIN ERRORS

## ABSTRACT

Diffusion or score-based models recently showed high performance in image generation. They rely on a forward and a backward stochastic differential equations (SDE). The sampling of a data distribution is achieved by solving numerically the backward SDE or its associated flow ODE. Studying the convergence of these models necessitates to control four different types of error: the initialization error, the truncation error, the discretization and the score approximation. In this paper, we study theoretically the behavior of diffusion models and their numerical implementation when the data distribution is Gaussian. In this restricted framework where the score function is a linear operator, we derive the analytical solutions of the backward SDE and the probability flow ODE. We prove that these solutions and their discretizations are all Gaussian processes, which allows us to compute exact Wasserstein errors induced by each error type for any sampling scheme. Monitoring convergence directly in the data space instead of relying on Inception features, our experiments show that the recommended numerical schemes from the diffusion models literature are also the best sampling schemes for Gaussian distributions.

## 1 INTRODUCTION

Over the last five years, diffusion models have proven to be a highly efficient and reliable framework for generative modeling (Song & Ermon, 2019; Ho et al., 2020; Song et al., 2021a;b; Dhariwal & Nichol, 2021; Karras et al., 2022). First introduced as a discrete process, Denoising Diffusion Probabilistic Models (DDPM) (Ho et al., 2020) can be studied as a reversal of a continuous Stochastic Differential Equation (SDE) (Song et al., 2021b). A forward SDE progressively transforms the initial data distribution by adding more and more noise as time grows. Then, the reversal of this process, called backward SDE, allows us to approximately sample the data distribution starting from Gaussian white noise. Moreover, the SDE is associated with an Ordinary Differential Equations (ODE) called probability flow (Song et al., 2021b). This flow preserves the same marginal distributions as the backward SDE and provides another way to sample the score-based generative model.

An important issue about diffusion models is the theoretical guarantees of convergence of the model: How close to the data distribution the generated distribution is? There are four main sources of errors to study for deriving theoretical guarantees for diffusion models: (a) the *initialization error* is induced when approximating the marginal distribution at the end of the forward process by a standard Gaussian distribution. (b) The *discretization error* comes from the resolution of the SDE or the ODE by a numerical method. (c) The *truncation error* occurs because the backward time integration is stopped at a small time $\epsilon > 0$ to avoid numerical instabilities due to ill-defined score function near the origin. (d) The *score approximation error* accounts for the mismatch between the ideal score function and the one given by the network trained using denoising score-matching.

Despite these numerous sources of errors, a lot of numerical and theoretical research has been led to assess the generative capacity of diffusion models. Several articles (Franzese et al., 2023; Karras et al., 2022) provide strong experimental studies for the choices of sampling parameters. On the theoretical side, several works derive upper bounds on the 1-Wasserstein or TV distance between the data and the model distributions by making assumptions on the $L^2$-error between the ideal and learned score functions and on the compacity of the support of the data (Chen et al., 2023b; Lee et al., 2024; De Bortoli et al., 2021; Chen et al., 2023c; Lee et al., 2022; Benton et al., 2024), eventually

under an additional manifold assumption (De Bortoli, 2022; Wenliang & Moran, 2022; Chen et al., 2023a). Yet, on one hand, to the best of our knowledge, the derived theoretical bounds mostly rely on worst case scenario and are not tight enough to explain the practical efficiency of diffusion models. On the other hand, numerical considerations mostly rely on Inception feature distributions through the FID metric (Heusel et al., 2017).

Ideally, given a data distribution of interest, one would like to have an adapted estimation of the discrepancy between the data and the diffusion model samples, thus enabling adaptive hyperparameter selection for the sampling procedure. As a first step towards reaching this goal, in the present work we study diffusion models applied to Gaussian data distributions. While this setting has a priori no practical interest, since simulating Gaussian variates does not require a diffusion model, it provides a large parametric family of distributions for which the errors involved in diffusion model sampling can be completely understood.

When restricting the data distribution to be Gaussian, the resulting score function is a simple linear operator. Exploiting this specificity allows us to derive the following contributions **under the assumption that the data distribution is Gaussian**:

- We give the exact solutions for both the backward SDE and the probability flow ODE.
- We fully describe the Gaussian processes that occur when using classical sampling discretization schemes.
- We derive exact 2-Wasserstein errors for the corresponding sample distributions and are able to assert for the influence of each error type on these errors, as illustrated by Figure 1.

Our theoretical study allows for an analytical evaluation of any numerical sampler, either stochastic or deterministic. In particular, it confirms the strength of best practice scheme such as Heun's method for the ODE flow (Karras et al., 2022). We provide our source code that can be applied to any Gaussian data distribution of interest and gives insight to calibrate parameters of a diffusion sampling algorithm, e.g. by straightforwardly generalizing our study to higher order linear numerical schemes.

While our theoretical analysis relies on an exactly known score function, we conduct additional experiments to assess for the influence of the score approximation error. Surprisingly, in the context of texture synthesis, we show that with a score neural network trained for modeling a specific Gaussian micro-texture a stochastic Euler-Maruyama sampler is more faithful to the data distribution than Heun's method, thus highlighting the importance of the score approximation error in practical situations.

**Plan of the paper:** First, we recall in Section 2 the continuous framework for SDE-based diffusion models. Section 3 presents our main theoretical results detailing the exact backward SDE and probability flow ODE solutions when supposing the data distribution to be Gaussian. Section 4 gives explicit Wasserstein error formulas when sampling the corresponding processes, yielding to an ablation study for comparing the influence of each error type on several sampling schemes. In Section 5, we study numerically a special case of Gaussian distribution for texture synthesis in order to evaluate the influence of the score approximation error occurring with a standard network architecture. Finally, we address discussion and limitations of our framework in Section 6.

## 2 PRELIMINARIES: SCORE-BASED MODELS THROUGH DIFFUSION SDES

This preliminary section follows the seminal work of Song *et al.* (Song et al., 2021b) and introduces specific notation to differentiate the exact backward process and the generative backward process obtained when starting from a white noise. Given a target distribution $p_{\text{data}}$ over $\mathbb{R}^d$, the forward diffusion process is the following variance preserving SDE

$$d\boldsymbol{x}_t = -\beta_t \boldsymbol{x}_t dt + \sqrt{2\beta_t} d\boldsymbol{w}_t, \quad 0 \leq t \leq T, \quad \boldsymbol{x}_0 \sim p_{\text{data}} \tag{1}$$

where $(\boldsymbol{w}_t)_{t \geq 0}$ is a $d$-dimensional Brownian motion and $\beta$ is a positive weight function. The distribution $p_{\text{data}}$ is noised progressively and the function $\beta$ is the variance of the added noise by time unit. We denote by $p_t$ the density of $(\boldsymbol{x}_t)$ for $t > 0$ since $p_{\text{data}}$ can be supported on a lower-dimensional manifold (De Bortoli, 2022). The SDE is designed so that $p_T$ is close to the Gaussian standard distribution that we denote $\mathcal{N}_0$ in whole paper. Under some assumptions on the distribution

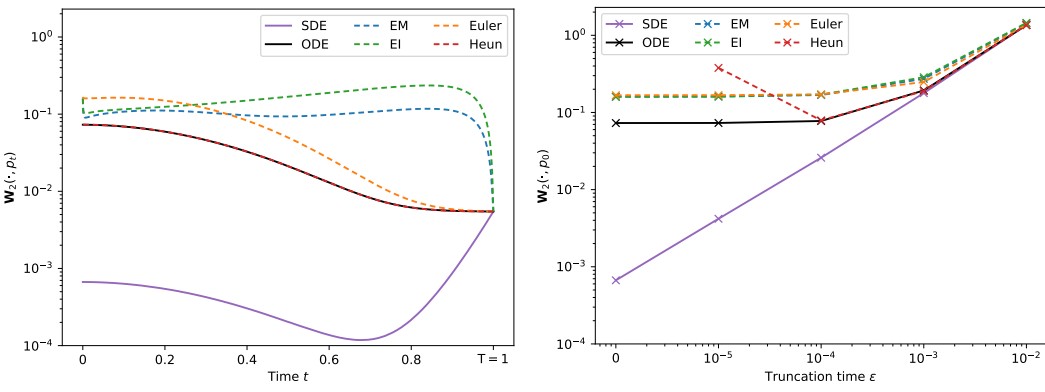

(a) Initialization error along the integration time

(b) Truncation error for different truncation time $\varepsilon$

Figure 1: **Wasserstein errors for the diffusion models associated with the CIFAR-10 Gaussian.** Left: Evolution of the Wasserstein distance between $p_t$ and the distributions associated with the continuous SDE, the continuous flow ODE and four discrete sampling schemes with standard $\mathcal{N}_0$ initialization, either stochastic (Euler-Maruyama (EM) and Exponential Integrator (EI)) or deterministic (Euler and Heun). While the continuous SDE is less sensible than the continuous ODE (as proved by Proposition 4), the initialization error impacts all discrete schemes with a comparable order of magnitude. Heun's method has the lowest error and is very close to the theoretical ODE, except for the last step (which is not represented) that is usually discarded when using time truncation. Right: Wasserstein errors due to time truncation for various truncation times $\epsilon$. Using time truncation increases the error for all the methods except Heun's scheme due to instability near the origin. Interestingly, for the standard practice truncation time $\varepsilon = 10^{-3}$, all numerical schemes have a comparable error close to their continuous counterparts.

$p_{\text{data}}$ (Pardoux, 1986), the backward process $(\boldsymbol{x}_{T-t})_{0 \le t \le T}$ verifies the backward SDE

$$d\boldsymbol{y}_t = \beta_{T-t}(\boldsymbol{y}_t + 2\nabla_{\boldsymbol{y}} \log p_{T-t}(\boldsymbol{y}_t))dt + \sqrt{2\beta_{T-t}}d\boldsymbol{w}_t, \quad 0 \le t < T, \quad \boldsymbol{y}_0 \sim p_T. \quad (2)$$

The objective is now to solve this reverse equation to sample $\boldsymbol{y}_T \sim p_{\text{data}}$. However, the distribution $p_T$ is in general not known, and image[1] generation is achieved by sampling

$$d\tilde{\boldsymbol{y}}_t = \beta_{T-t}(\tilde{\boldsymbol{y}}_t + 2\nabla_{\boldsymbol{y}} \log p_{T-t}(\tilde{\boldsymbol{y}}_t))dt + \sqrt{2\beta_{T-t}}d\boldsymbol{w}_t, \quad 0 \le t < T, \quad \tilde{\boldsymbol{y}}_0 \sim \mathcal{N}_0. \quad (3)$$

Note that approximating $p_T$ by $\mathcal{N}_0$ for the initialization $\boldsymbol{y}_0$ makes that the solution of the SDE of Equation (3) is not exactly the target distribution $p_{\text{data}}$. An alternative way to approximately sample $p_{\text{data}}$ is to use that every diffusion process is associated with a deterministic process whose trajectories share the same marginal probability densities $(p_t)_{0 < t \le T}$ as the SDE (Song et al., 2021b). The deterministic process associated with Equation (2) is

$$d\boldsymbol{x}_t = [-\beta_t \boldsymbol{x}_t - \beta_t \nabla_{\boldsymbol{x}} \log p_t(\boldsymbol{x}_t)]\, dt, \quad 0 < t \le T, \quad \boldsymbol{x}_0 \sim p_{\text{data}}. \quad (4)$$

This ODE can be solved in reverse-time to sample $\boldsymbol{x}_0$ from $\boldsymbol{x}_T \sim p_T$. Given $(\boldsymbol{x}_t)_{0 \le t \le T}$ solution of Equation (4), $(\boldsymbol{x}_{T-t})_{0 \le t \le T}$ is solution of

$$d\boldsymbol{y}_t = [\beta_{T-t} \boldsymbol{y}_t + \beta_{T-t} \nabla_{\boldsymbol{y}} \log p_{T-t}(\boldsymbol{y}_t)]\, dt, \quad 0 \le t < T. \quad (5)$$

Again, in practice, the ODE which is considered to achieve image generation is

$$d\widehat{\boldsymbol{y}}_t = [\beta_{T-t} \widehat{\boldsymbol{y}}_t + \beta_{T-t} \nabla_{\widehat{\boldsymbol{y}}} \log p_{T-t}(\widehat{\boldsymbol{y}}_t)]\, dt, \quad 0 \le t < T, \quad \widehat{\boldsymbol{y}}_0 \sim \mathcal{N}_0, \quad (6)$$

where $p_T$ is replaced by $\mathcal{N}_0$. As a consequence of this approximation, the property of conservation of the marginals $(p_t)_{0 \le t \le T}$ does not occur. We denote by $(\tilde{q}_t)_{0 \le t \le T}$, respectively $(\widehat{q}_t)_{0 \le t \le T}$, the marginals of $(\tilde{\boldsymbol{y}}_t)_{0 \le t \le T}$ and $(\widehat{\boldsymbol{y}}_t)_{0 \le t \le T}$ and $\tilde{p}_t = \tilde{q}_{T-t}$, $\widehat{p}_t = \widehat{q}_{T-t}$ the marginals of $(\tilde{\boldsymbol{y}}_{T-t})_{0 \le t \le T}$ and $(\widehat{\boldsymbol{y}}_{T-t})_{0 \le t \le T}$ such that $\tilde{p}_t$ and $\widehat{p}_t$ are approximations of $p_t$.

---

[1]Although we may refer to data as images, our analysis is fully general and applies to any vector-valued diffusion model.

## 3 EXACT SDE AND ODE SOLUTIONS

Our approach relies on deriving explicit solutions to the various SDE and ODE. We begin with the forward SDE in full generality obtained in applying the variation of constants (see the proof in Appendix B.1). This resolution also provides an ODE verified by the covariance matrix of $\boldsymbol{x}_t$, that we denote $\boldsymbol{\Sigma}_t = \mathrm{Cov}(\boldsymbol{x}_t)$.

**Proposition 1** (Solution of the forward SDE). *The strong solution of Equation* (1) *can be written as:*

$$\boldsymbol{x}_t = e^{-B_t}\boldsymbol{x}_0 + \boldsymbol{\eta}_t, \quad 0 \le t \le T, \tag{7}$$

*where $B_t = \int_0^t \beta_s ds$ and $\boldsymbol{\eta}_t = e^{-B_t}\int_0^t e^{B_s}\sqrt{2\beta_s}d\boldsymbol{w}_s$ is a Gaussian process independent of $\boldsymbol{x}_0$ whose covariance matrix is $(1 - e^{-2B_t})\boldsymbol{I}$. Consequently, the covariance matrix $\boldsymbol{\Sigma}_t$ of $\boldsymbol{x}_t$ is*

$$\boldsymbol{\Sigma}_t = e^{-2B_t}\boldsymbol{\Sigma} + (1 - e^{-2B_t})\boldsymbol{I}. \tag{8}$$

*where $\boldsymbol{\Sigma}$ is the covariance matrix of $\boldsymbol{x}_0 \sim p_{\mathrm{data}}$. Futhermore, $\boldsymbol{\Sigma}_t$ is invertible for $t > 0$ and verifies the matrix-valued ODE*

$$d\boldsymbol{\Sigma}_t = 2\beta_t(\boldsymbol{I} - \boldsymbol{\Sigma}_t)dt, \quad 0 < t \le T. \tag{9}$$

For a general data distribution $p_{\mathrm{data}}$, solving the backward SDE in infeasible, the main reason being that the expression of the score function to integrate is unknown. To circumvent this obstacle, we now suppose that the data distribution is Gaussian.

**Assumption 1** (Gaussian assumption). *$p_{\mathrm{data}}$ is a centered Gaussian distribution $\mathcal{N}(\boldsymbol{0}, \boldsymbol{\Sigma})$.*

Note that $\boldsymbol{\Sigma}$ may be non-invertible and thus $p_{\mathrm{data}}$ supported on a strict subspace of $\mathbb{R}^d$, a special case of manifold hypothesis. Consequently, the matrix $\boldsymbol{\Sigma}_t$ is in general only invertible for $t > 0$. Under Gaussian assumption, $(\boldsymbol{x}_t)$ is a Gaussian process with marginal distribution $p_t = \mathcal{N}(\boldsymbol{0}, \boldsymbol{\Sigma}_t)$ and consequently the score is the linear function

$$\nabla \log p_t(\boldsymbol{x}) = -\boldsymbol{\Sigma}_t^{-1}\boldsymbol{x}, \quad 0 < t \le T. \tag{10}$$

Note that the linearity of the diffusion score characterizes Gaussian distributions as detailed by Proposition 5 in Appendix A.

The cornerstone of our work is that under Gaussian assumption we can derive an exact solution of the backward SDE, without supposing that the initial condition is Gaussian.

**Proposition 2** (Solution of the backward SDE under Gaussian assumption). *Under Gaussian assumption, the strong solution to the SDE of Equation* (2)*:*

$$d\boldsymbol{y}_t = \beta_{T-t}(\boldsymbol{y}_t + 2\nabla_{\boldsymbol{y}} \log p_{T-t}(\boldsymbol{y}_t))dt + \sqrt{2\beta_{T-t}}d\boldsymbol{w}_t, \quad 0 \le t < T \tag{11}$$

*with $\boldsymbol{y}_0$ following any initial distribution can be written as:*

$$\boldsymbol{y}_t = e^{-(B_T - B_{T-t})}\boldsymbol{\Sigma}_{T-t}\boldsymbol{\Sigma}_T^{-1}\boldsymbol{y}_0 + \boldsymbol{\xi}_t, \quad 0 \le t \le T \tag{12}$$

*where $\boldsymbol{\xi}_t = e^{-(B_T - B_{T-t})}\boldsymbol{\Sigma}_{T-t}\int_0^t \boldsymbol{\Sigma}_{T-s}^{-1}e^{-(B_T - B_{T-s})}\sqrt{2\beta_{T-s}}d\boldsymbol{w}_s$ is a Gaussian process with covariance matrix $\mathrm{Cov}(\boldsymbol{\xi}_t) = \boldsymbol{\Sigma}_{T-t} - e^{-2(B_T - B_{T-t})}\boldsymbol{\Sigma}_{T-t}^2\boldsymbol{\Sigma}_T^{-1}$. Finally:*

$$\mathrm{Cov}(\boldsymbol{y}_t) = \boldsymbol{\Sigma}_{T-t} + e^{-2(B_T - B_{T-t})}\boldsymbol{\Sigma}_{T-t}^2\boldsymbol{\Sigma}_T^{-1}\left(\boldsymbol{\Sigma}_{T-t}^{-1}\mathrm{Cov}(\boldsymbol{y}_0)\boldsymbol{\Sigma}_T^{-1}\boldsymbol{\Sigma}_{T-t} - \boldsymbol{I}\right), \tag{13}$$

*and in particular, if $\mathrm{Cov}(\boldsymbol{y}_0)$ and $\boldsymbol{\Sigma}$ commute,*

$$\mathrm{Cov}(\boldsymbol{y}_t) = \boldsymbol{\Sigma}_{T-t} + e^{-2(B_T - B_{T-t})}\boldsymbol{\Sigma}_{T-t}^2\boldsymbol{\Sigma}_T^{-2}\left[\mathrm{Cov}(\boldsymbol{y}_0) - \boldsymbol{\Sigma}_T\right]. \tag{14}$$

While not as straightforward as the forward case, the proof also relies on applying the variation of constants and is given in Appendix B.2. Note that if $\boldsymbol{y}_0$ is correctly initialized at $p_T$, $y_{T-t} \sim p_t$ at each time $0 \le t \le T$. As shown by the following proposition (proved in Appendix B.3), the flow ODE also has an explicit solution under Gaussian assumption which is related to optimal transport (OT).

**Proposition 3** (Solution of the ODE probability flow under Gaussian assumption). *The solution to the reverse-time probability flow ODE of Equation* (5):

$$d\boldsymbol{y}_t = [\beta_{T-t}\boldsymbol{y}_t + \beta_{T-t}\nabla_{\boldsymbol{y}}\log p_{T-t}(\boldsymbol{y}_t)]\, dt, \quad 0 \le t < T \tag{15}$$

*for any $\boldsymbol{y}_0$ is:*

$$\boldsymbol{y}_t = \boldsymbol{\Sigma}_T^{-1/2}\boldsymbol{\Sigma}_{T-t}^{1/2}\boldsymbol{y}_0, \quad 0 \le t \le T, \tag{16}$$

*which is the application of the OT map between $p_T$ and $p_{T-t}$ to the initial condition $\boldsymbol{y}_0$. Consequently, the covariance matrix $\mathrm{Cov}(\boldsymbol{y}_t)$ verifies*

$$\mathrm{Cov}(\boldsymbol{y}_t) = \boldsymbol{\Sigma}_T^{-1/2}\boldsymbol{\Sigma}_{T-t}^{1/2}\mathrm{Cov}(\boldsymbol{y}_0)\boldsymbol{\Sigma}_{T-t}^{1/2}\boldsymbol{\Sigma}_T^{-1/2}, \quad 0 \le t \le T, \tag{17}$$

*and in particular, if $\mathrm{Cov}(\boldsymbol{y}_0)$ and $\boldsymbol{\Sigma}$ commute,*

$$\mathrm{Cov}(\boldsymbol{y}_t) = \boldsymbol{\Sigma}_{T-t}\boldsymbol{\Sigma}_T^{-1}\mathrm{Cov}(\boldsymbol{y}_0), \quad 0 \le t \le T. \tag{18}$$

Here we must highlight a subtle issue: Whatever the initial distribution of $\boldsymbol{y}_0$ is, the ODE solution consists in applying the OT map between $p_T$ and $p_{T-t}$ at time $t$. If $\boldsymbol{y}_0$ follows $p_T$, then $\boldsymbol{y}_{T-t} \sim p_t$ at each time $0 \le t \le T$. But since in practice one cannot truly sample $p_T$ and uses $\boldsymbol{y}_0 \sim \mathcal{N}_0$ instead, the resulting flow is not an OT flow (even though it involves an OT mapping) and the distribution of $\boldsymbol{y}_T$ differs from $p_{\text{data}}$.

**Links with related work.** Some parts of Propositions 1, 2 and 3 have been stated in previous work. Equation (7) of Proposition 1 is given without proof in (Gao & Zhu, 2024), the variance ODE, that we generalize here to the full covariance matrix (Equation (9)), is given in (Song et al., 2021b), (Särkkä & Solin, 2019, Equation 6.20)), and the score expression under Gaussian assumption is reported in several recent references Albergo et al. (2023); Zach et al. (2024; 2023); Shah et al. (2023).

To the best of our knowledge Proposition 2 is new and is the cornerstone for our analytical and numerical study. Gaussian mixtures have been studied in the context of diffusion models (Zach et al., 2024; 2023; Shah et al., 2023) since they also provide an explicit analytical score. However, solving exactly the backward SDE is not feasible for Gaussian mixtures as far as we know.

The relation between OT and probability flow ODE has been discussed in (Lavenant & Santambrogio, 2022; Khrulkov et al., 2023). Lavenant & Santambrogio (2022) show that, in general, the flow ODE solution is not an OT between $p_{\text{data}}$ and $\mathcal{N}_0$ at infinite time $T \to +\infty$, thus contradicting a conjecture of Khrulkov et al. (2023). Yet, they briefly discuss the Gaussian case as special case for which the conjecture is valid. Indeed, Khrulkov et al. (2023) derive the solution of the flow ODE under Gaussian assumption at infinite time horizon (Khrulkov et al., 2023, Appendix B). More recently, an expression of the solution of the flow ODE relying on the eigendecomposition of the covariance matrix of the data in Gaussian case is given in (Wang & Vastola, 2023) assuming $\boldsymbol{y}_0 \sim \mathcal{N}_0$. None of these works discuss the mismatch between the OT map and the initialization of $\boldsymbol{y}_0$. Our Proposition 3 highlights that the generated process is not an OT flow due to the initialization error.

## 4 EXACT WASSERSTEIN ERRORS

The specificity of the Gaussian case allows us to study precisely the different types of error with the expression of the explicit solution of the backward SDE. In what follows, we designate by Wasserstein distance the 2-Wasserstein distance which is known in closed forms when applied to Gaussian distributions (Dowson & Landau, 1982). For two centered Gaussians $\mathcal{N}(\boldsymbol{0}, \boldsymbol{\Sigma}_1)$ and $\mathcal{N}(\boldsymbol{0}, \boldsymbol{\Sigma}_2)$ such that $\boldsymbol{\Sigma}_1, \boldsymbol{\Sigma}_2$ are simultaneously diagonalizable with respective eigenvalues $(\lambda_{i,1})_{1 \le i \le d}, (\lambda_{i,2})_{1 \le i \le d}$,

$$\mathbf{W}_2(\mathcal{N}(\boldsymbol{0}, \boldsymbol{\Sigma}_1), \mathcal{N}(\boldsymbol{0}, \boldsymbol{\Sigma}_2))^2 = \sum_{1 \le i \le d} (\sqrt{\lambda_{i,1}} - \sqrt{\lambda_{i,2}})^2 \tag{19}$$

as used in (Ferradans et al., 2013). In the literature, the quality of the diffusion models is measured with FID (Heusel et al., 2017) which is the $\mathbf{W}_2$-error between Gaussians fitted to the Inception features (Szegedy et al., 2016) of two discrete datasets. Here we use the $\mathbf{W}_2$-errors directly in data space, which is more informative and allows us to provide theoretical $\mathbf{W}_2$-errors. To illustrate our theoretical results, we consider the CIFAR-10 Gaussian distribution, that is, the Gaussian distribution such that $\boldsymbol{\Sigma}$ is the empirical covariance of the CIFAR-10 dataset. As shown in Appendix C, images produced by this model are not interesting due to a lack of structure, but the corresponding covariance has the advantage of reflecting the complexity of real data.

**The initialization error.** As discussed in Sections 2 and 3, the marginals of both generative processes $\tilde{y}$ and $\hat{y}$ following respectively Equation (6) and Equation (3) slightly differs from $p_t$ due to their common white noise initial condition. This implies an error that we call the initialization error. The distance between $(\tilde{p}_t)_{0 \leq t \leq T}$, $(\hat{p}_t)_{0 \leq t \leq T}$ and $(p_t)_{0 \leq t \leq T}$ can be explicitly studied in the Gaussian case with the following proposition (proved in Appendix B.4).

**Proposition 4** (Marginals of the generative processes under Gaussian assumption). *Under Gaussian assumption, $(\tilde{y}_t)_{0 \leq t \leq T}$ and $(\hat{y}_t)_{0 \leq t \leq T}$ are Gaussian processes. At each time $t$, $\tilde{p}_t$ is the Gaussian distribution $\mathcal{N}(\mathbf{0}, \tilde{\boldsymbol{\Sigma}}_t)$ with $\tilde{\boldsymbol{\Sigma}}_t = \boldsymbol{\Sigma}_t + e^{-2(B_T - B_t)} \boldsymbol{\Sigma}_t^2 \boldsymbol{\Sigma}_T^{-1} (\boldsymbol{\Sigma}_T^{-1} - \boldsymbol{I})$ and $\hat{p}_t$ is the Gaussian distribution $\mathcal{N}(\mathbf{0}, \hat{\boldsymbol{\Sigma}}_t)$ with $\hat{\boldsymbol{\Sigma}}_t = \boldsymbol{\Sigma}_T^{-1} \boldsymbol{\Sigma}_t$. For all $0 \leq t \leq T$, the three covariance matrices $\boldsymbol{\Sigma}_t$, $\tilde{\boldsymbol{\Sigma}}_t$ and $\hat{\boldsymbol{\Sigma}}_t$ share the same range. Furthermore, for all $0 \leq t \leq T$,*

$$\boldsymbol{W}_2(\tilde{p}_t, p_t) \leq \boldsymbol{W}_2(\hat{p}_t, p_t) \tag{20}$$

*which shows that at each time $0 \leq t \leq T$ and in particular for $t = 0$ which corresponds to the desired outputs of the sampler, the SDE sampler is a better sampler than the ODE sampler when the exact score is konwn.*

In practice the initialization error for the SDE and ODE samplers may vary by several orders of magnitude, as shown for the CIFAR-10 example in Figure 1.(a) (solid lines) which illustrates Equation (20).

**The discretization error.** The implementation of the SDE and the ODE implies to choose a discrete numerical scheme. We propose to study four different schemes presented in Table 1. The classical Euler-Maruyama (EM) is used in (Song et al., 2021b) and the exponential integrator (EI) in (De Bortoli, 2022) to sample from the SDE of Equation (3). The Euler method is the simplest ODE solver and Heun's scheme is recommended in (Karras et al., 2022) to model the ODE of Equation (6). Under Gaussian assumption, the eigenvalues of the covariance matrices can be computed numerically recursively for each scheme to evaluate the Wasserstein distance. Indeed, under Gaussian assumption, the score is a linear operator and the discrete schemes lead to linear operations described in Table 1. Then, a Gaussian initialization for $\boldsymbol{y}_0$ provides a sequence of centered Gaussian processes $(y_k^{\Delta,\cdot})_k$ and if $\boldsymbol{y}_0$ follows $p_T$ or $\mathcal{N}_0$, the covariance matrix $\text{Cov}(y_k^{\Delta,\cdot})$ admit the same eigenvectors as $\boldsymbol{\Sigma}$ and we can use Equation (19) to compute Wasserstein distances. Let us illustrate the computation of the eigenvalues with the EM scheme. Denoting $(\lambda_{i,t})_{1 \leq i \leq d}$ the eigenvalues of $\boldsymbol{\Sigma}_t$ and $\left(\lambda_{i,k}^{\Delta,\text{EM}}\right)_{1 \leq i \leq d}$ the eigenvalues of the covariance matrix of the Euler-Maruyama discretization of the SDE at the $k$th step, $1 \leq k \leq N - 1$, the relation verified by these eigenvalues is

$$\lambda_{i,k+1}^{\Delta,\text{EM}} = \left(1 + \Delta_t \beta_{T-t_k}(1 - \tfrac{2}{\lambda_{i,T-t_k}})\right)^2 \lambda_{i,k}^{\Delta,\text{EM}} + 2\Delta_t \beta_{T-t_k}, 1 \leq i \leq d, 0 \leq k \leq N - 2 \tag{21}$$

with initialization $\lambda_{i,0}^{\Delta,\text{EM}} = \begin{cases} 1 & \text{if } \boldsymbol{y}_T \text{ is initialized at } \mathcal{N}_0 \\ \lambda_{i,T} & \text{if } \boldsymbol{y}_T \text{ is initialized at } p_T \end{cases}$ $1 \leq i \leq d$.. More detailed computations for EM and formulas for other schemes are presented in Appendix D. For each scheme, we recursively compute the eigenvalues at each time discretization and present the observed Wasserstein distance in Figure 1.(a). We can observe that Heun's method provide the lower Wasserstein distance, followed by EM, EI and the Euler scheme. Note that the discrete schemes does not preserve the range of the covariance matrix, contrary to the continuous formulas. This explains the fact that the Wasserstein distance increases at the final step.

**The truncation error.** As discussed in (Song et al., 2021b), it is preferable to study the backward process on $[\varepsilon, T]$ instead of $[0, T]$ because the score is a priori not defined for $t = 0$, which occurs in our case if $\boldsymbol{\Sigma}$ is not invertible. This approximation is called the truncation error. As a consequence, even without initialization error, the backward process leads to $p_\varepsilon$ and not $p_0$. Under Gaussian assumption, it is possible to explicit this error with the expression given in Proposition 3 and 2 as done in Figure 1.(b) for both continuous and numerical solutions. For the standard practice truncation time $\varepsilon = 10^{-3}$ (Song et al., 2021b; Karras et al., 2022), all numerical schemes have an error close to the corresponding continuous solution. Using a lower $\varepsilon$ value is only relevant for the continuous SDE solution.

| | | |
|---|---|---|
| SDE schemes | Euler-Maruyama (EM) | $\begin{cases} \tilde{\boldsymbol{y}}_0^{\Delta,\text{EM}} & \sim \mathcal{N}_0 \\ \tilde{\boldsymbol{y}}_{k+1}^{\Delta,\text{EM}} & = \tilde{\boldsymbol{y}}_k^{\Delta,\text{EM}} + \Delta_t \beta_{T-t_k} \left( \tilde{\boldsymbol{y}}_k^{\Delta,\text{EM}} - 2\boldsymbol{\Sigma}_{T-t_k}^{-1} \tilde{\boldsymbol{y}}_k^{\Delta,\text{EM}} \right) + \sqrt{2\Delta_t \beta_{T-t_k}} \boldsymbol{z}_k, \; \boldsymbol{z}_k \sim \mathcal{N}_0 \end{cases}$ (22) |
| | Exponential integrator (EI) | $\begin{cases} \tilde{\boldsymbol{y}}_0^{\Delta,\text{EI}} & \sim \mathcal{N}_0 \\ \tilde{\boldsymbol{y}}_{k+1}^{\Delta,\text{EI}} & = \tilde{\boldsymbol{y}}_k^{\Delta,\text{EI}} + \gamma_{1,k} \left( \tilde{\boldsymbol{y}}_k^{\Delta,\text{EI}} - 2\boldsymbol{\Sigma}_{T-t_k}^{-1} \tilde{\boldsymbol{y}}_k^{\Delta,\text{EI}} \right) + \sqrt{2\gamma_{2,k}} \boldsymbol{z}_k, \; \boldsymbol{z}_k \sim \mathcal{N}_0 \\ \text{where } \gamma_{1,k} = \exp(B_{T-t_k} - B_{T-t_{k+1}}) - 1 \text{ and } \gamma_{2,k} = \frac{1}{2}(\exp(2B_{T-t_k} - 2B_{T-t_{k+1}}) - 1) \end{cases}$ (23) |
| ODE schemes | Explicit Euler | $\begin{cases} \widehat{\boldsymbol{y}}_0^{\Delta,\text{Euler}} & \sim \mathcal{N}_0 \\ \widehat{\boldsymbol{y}}_{k+1}^{\Delta,\text{Euler}} & = \widehat{\boldsymbol{y}}_k^{\Delta,\text{Euler}} + \Delta_t f(t_k, \widehat{\boldsymbol{y}}_k^{\Delta,\text{Euler}}) \quad \text{with } f(t,\boldsymbol{y}) = \beta_{T-t} \boldsymbol{y} - \beta_{T-t} \boldsymbol{\Sigma}_{T-t}^{-1} \boldsymbol{y} \end{cases}$ (24) |
| | Heun's method | $\begin{cases} \widehat{\boldsymbol{y}}_0^{\Delta,\text{Heun}} & \sim \mathcal{N}_0 \\ \widehat{\boldsymbol{y}}_{k+1/2}^{\Delta,\text{Heun}} & = \widehat{\boldsymbol{y}}_k^{\Delta,\text{Heun}} + \Delta_t f(t_k, \widehat{\boldsymbol{y}}_k^{\Delta,\text{Heun}}) \quad \text{with } f(t,\boldsymbol{y}) = \beta_{T-t} \boldsymbol{y} - \beta_{T-t} \boldsymbol{\Sigma}_{T-t}^{-1} \boldsymbol{y} \\ \widehat{\boldsymbol{y}}_{k+1}^{\Delta,\text{Heun}} & = \widehat{\boldsymbol{y}}_k^{\Delta,\text{Heun}} + \frac{\Delta_t}{2} \left( f(t_k, \widehat{\boldsymbol{y}}_k^{\Delta,\text{Heun}}) + f(t_{k+1}, \widehat{\boldsymbol{y}}_{k+1/2}^{\Delta,\text{Heun}}) \right) \end{cases}$ (25) |

Table 1: **Stochastic and deterministic discretization schemes**. EM and EI disctretize the backward SDE of Equation (3), Euler and Heun schemes discretize of the probability flow ODE of Equation (6) with a regular time schedule $(t_k)_{0 \le k \le N}$ with stepsize $\Delta_t = \frac{T}{N}$.

| | | Continuous | | $N = 50$ | | $N = 250$ | | $N = 500$ | | $N = 1000$ | |
|---|---|---|---|---|---|---|---|---|---|---|---|
| | | $p_T$ | $\mathcal{N}_0$ | $p_T$ | $\mathcal{N}_0$ | $p_T$ | $\mathcal{N}_0$ | $p_T$ | $\mathcal{N}_0$ | $p_T$ | $\mathcal{N}_0$ |
| EM | $\varepsilon = 0$ | 0 | 6.7E-04 | 4.78 | 4.78 | 0.65 | 0.66 | 0.32 | 0.32 | 0.16 | 0.16 |
| | $\varepsilon = 10^{-5}$ | 4.1E-03 | 4.2E-03 | 4.77 | 4.77 | 0.66 | 0.66 | 0.32 | 0.32 | 0.16 | 0.16 |
| | $\varepsilon = 10^{-4}$ | 0.03 | 0.03 | 4.76 | 4.76 | 0.66 | 0.66 | 0.32 | 0.32 | 0.17 | 0.17 |
| | $\varepsilon = 10^{-3}$ | 0.18 | 0.18 | 4.68 | 4.68 | 0.70 | 0.70 | 0.40 | 0.40 | 0.27 | 0.27 |
| EI | $\varepsilon = 0$ | 0 | 6.7E-04 | 2.81 | 2.81 | 0.57 | 0.57 | 0.30 | 0.30 | 0.16 | 0.16 |
| | $\varepsilon = 10^{-5}$ | 4.1E-03 | 4.2E-03 | 2.81 | 2.81 | 0.57 | 0.57 | 0.30 | 0.30 | 0.16 | 0.16 |
| | $\varepsilon = 10^{-4}$ | 0.03 | 0.03 | 2.82 | 2.82 | 0.58 | 0.58 | 0.31 | 0.31 | 0.17 | 0.17 |
| | $\varepsilon = 10^{-3}$ | 0.18 | 0.18 | 2.91 | 2.91 | 0.67 | 0.67 | 0.41 | 0.41 | 0.29 | 0.29 |
| Euler | $\varepsilon = 0$ | 0 | 0.07 | 1.72 | 1.78 | 0.38 | 0.44 | 0.20 | 0.26 | 0.10 | 0.17 |
| | $\varepsilon = 10^{-5}$ | 4.1E-03 | 0.07 | 1.72 | 1.78 | 0.38 | 0.44 | 0.20 | 0.26 | 0.10 | 0.17 |
| | $\varepsilon = 10^{-4}$ | 0.03 | 0.08 | 1.72 | 1.78 | 0.38 | 0.44 | 0.20 | 0.26 | 0.11 | 0.17 |
| | $\varepsilon = 10^{-3}$ | 0.18 | 0.19 | 1.73 | 1.79 | 0.42 | 0.48 | 0.27 | 0.32 | 0.21 | 0.25 |
| Heun | $\varepsilon = 0$ | 0 | 0.07 | - | - | - | - | - | - | - | - |
| | $\varepsilon = 10^{-5}$ | 4.1E-03 | 0.07 | 23.42 | 23.42 | 2.86 | 2.87 | 1.05 | 1.06 | 0.37 | 0.38 |
| | $\varepsilon = 10^{-4}$ | 0.03 | 0.08 | 4.68 | 4.68 | 0.43 | 0.44 | 0.12 | 0.14 | 0.03 | 0.08 |
| | $\varepsilon = 10^{-3}$ | 0.18 | 0.19 | 0.58 | 0.59 | 0.13 | 0.15 | 0.16 | 0.18 | 0.17 | 0.19 |

Table 2: **Ablation study of Wasserstein errors for the CIFAR-10 Gaussian.** For a given discretization scheme, the table presents the Wasserstein distance associated with the truncation error for different values of $\varepsilon$. The columns $p_T$ and $\mathcal{N}_0$ show the influence of the initialization error. The continuous column corresponds to the continuous SDE or ODE linked with the scheme (identical values for EM, EI and Euler, Heun) and a given number of integration steps $N$.

**Ablation study.** We propose in Table 2 an ablation study to monitor the magnitude of each error and their accumulation for various sampling schemes for the CIFAR-10 example. In accordance with Proposition 4, the initialization error influences the ODE schemes, while SDE schemes are not affected. Schemes having a sufficient number of steps are not sensitive to the truncation error for $\varepsilon < 10^{-3}$, except Heun's scheme which is unstable near to origin. The discretization error is the more important approximation but it becomes very low for a sufficient number of steps. The lower Wasserstein error is provided by Heun's method with 1000 steps, $\varepsilon = 10^{-4}$. As Karras et al. (2022), our conclusions lead to the choice of Heun's scheme as the go-to method.

**Influence of eigenvalues.** The above observations and conclusions are observed on the CIFAR-10 Gaussian. However, in general, they depend on the eigenvalues of the covariance matrix $\boldsymbol{\Sigma}$. Indeed, as seen in Equation (19), the Wasserstein distance is separable and each eigenvalue contributes to increase it. In Figure 2, we evaluate the contribution of each eigenvalue by plotting $\lambda \mapsto |\sqrt{\lambda} - \sqrt{\lambda^{\text{scheme}}}|$ for

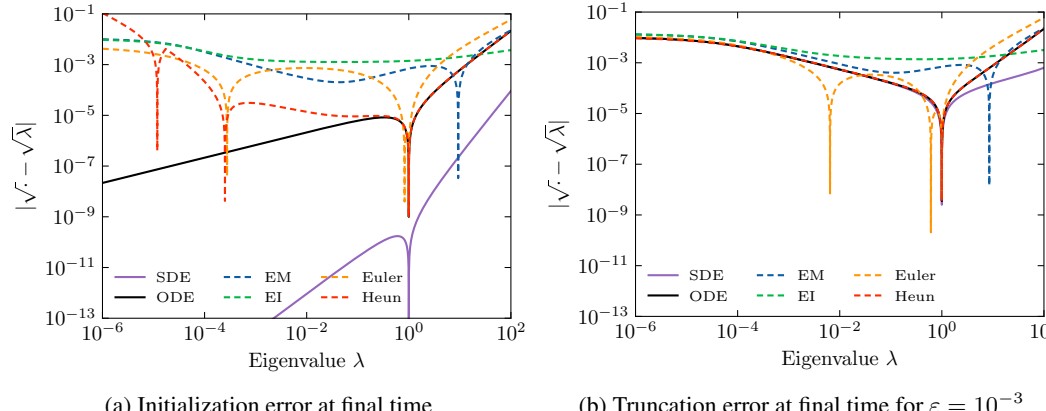

(a) Initialization error at final time

(b) Truncation error at final time for $\varepsilon = 10^{-3}$

Figure 2: **Eigenvalue contribution to the Wasserstein error.** The magnitude of the Wasserstein error is influenced by the eigenvalues of the covariance of the Gaussian distribution. Left: Contribution to the Wasserstein error for the continuous equations and the discretization schemes with standard initialization $\mathcal{N}_0$. Right: Same plot when using a truncation time $\varepsilon = 10^{-3}$. All schemes use $N = 1000$ steps. While we prove that the continuous SDE is always better than the continuous ODE (Proposition 4), it is not the same for the discrete schemes. With a truncation time $\varepsilon = 10^{-3}$ (b), Heun's method is nearly as good as the continuous ODE solution for all eigenvalues, which shows it is well-adapted to any Gaussian distribution.

each scheme. Figure 2.(a) demonstrates that for the continuous equations, the error increases with the eigenvalues except for a strong decrease for $\lambda = 1$. Besides, as proved in the proof of Proposition 4 (see Appendix B.4), the error for the SDE is always lower than the error for the ODE and we can observe how tight is Equation (20). Unfortunately, once discretized the stochastic schemes are not as good as the continuous solutions. The EI scheme is the more stable along the range of eigenvalues but in the end it is in general more costly than the others in terms of Wasserstein error. Without truncation time, Heun's method fails for low eigenvalues because $\Sigma$ is not stably invertible. However, as seen in Figure 2.(b), with a truncation time $\varepsilon = 10^{-3}$, Heun's method is very close to the continuous ODE solution. This shows that for any Gaussian distribution Heun's method introduces nearly no additional discretization error, making this scheme the one to favor in practice. Our code allows for the evaluation of any covariance matrix and the computation of Figure 1 and Table 2 (provided the eigenvalues can be computed, see supp. mat.).

## 5 NUMERICAL STUDY OF THE SCORE APPROXIMATION

So far our theoretical and numerical study has been conducted under the hypothesis that the score function is known, thus discarding the evaluation of the score approximation. In practice, for general data distribution, the score function is parameterized by a neural network trained using denoising score-matching. This learned score function is not perfect and while theoretical studies assume the network to be close to the theoretical one (with uniform or adaptative bounds, see the discussion in (De Bortoli, 2022)), such an hypothesis is hard to check in practice, especially in our non compact setting. Thus, we propose in this section to train a diffusion models on a Gaussian distribution and evaluate numerically the impact of the score approximation.

**The Gaussian ADSN distribution for microtextures.** So far our running example was the CIFAR-10 Gaussian but we will now turn to another example that produces visually interesting images, namely Gaussian micro-textures. We consider the asymptotic discrete spot noise (ADSN) distribution (Galerne et al., 2011) associated with an RGB texture $\boldsymbol{u} \in \mathbb{R}^{3 \times M \times N}$ which is defined as the stationary Gaussian distribution that has covariance equal the autocorrelation of $\boldsymbol{u}$. More precisely, this distribution is sampled using convolution with a white Gaussian noise (Galerne et al., 2011): Denoting $m \in \mathbb{R}^3$ the channelwise mean of $\boldsymbol{u}$ and $\mathbf{t}_c = \frac{1}{\sqrt{MN}}(\boldsymbol{u}_c - m_c)$, $1 \le c \le 3$, its associated *texton*, for $\boldsymbol{w} \sim \mathcal{N}_0$ of size $M \times N$ the channelwise convolution $\boldsymbol{x} = m + \mathbf{t} \star \boldsymbol{w} \in \mathbb{R}^{3 \times M \times N}$ follows ADSN($\boldsymbol{u}$). This distribution is the Gaussian $\mathcal{N}(m, \boldsymbol{\Sigma})$. To deal with zero mean Gaussian, adding the mean $m$ is considered as a post-processing to visualize samples and we study $\mathcal{N}(\mathbf{0}, \boldsymbol{\Sigma})$. The

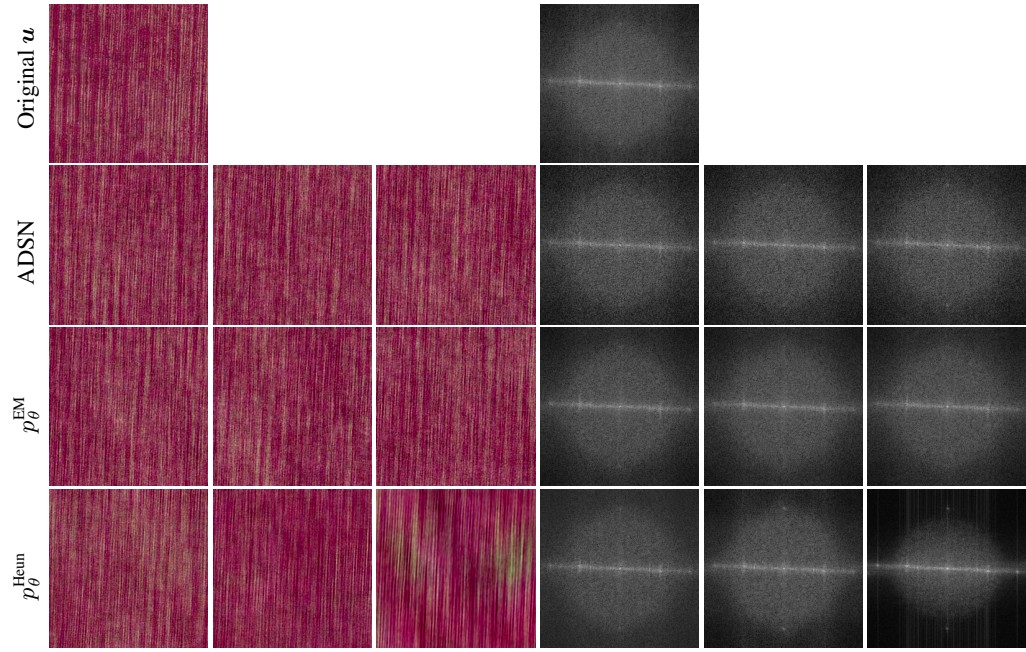

Figure 3: **Texture samples generated with the learned score.** First row: original image $\boldsymbol{u}$ and its DFT modulus (for all DFT modulus we display the sum of the DFT modulus of the three color channels and apply a logarithmic contrast change). Second row: three samples of $\mathrm{ADSN}(\boldsymbol{u})$ with their associated DFT moduli. Third and fourth row: Samples generated with the learned score with EM and Heun's discretization schemes and their associated DFT moduli. While both schemes use the same learned score function, the generation with Heun's scheme can produce out-of-distribution samples, as seen with the third sample.

matrix $\boldsymbol{\Sigma}$ is a well-known convolution matrix (Ferradans et al., 2013), its eigenvectors and associated eigenvalues can be computed in the Fourier domain, as done in Appendix F.2. $\boldsymbol{\Sigma}$ admits the eigenvalues $\lambda_1^{\xi,\mathrm{ADSN}} = |\widehat{\mathbf{t}}_1|^2(\xi) + |\widehat{\mathbf{t}}_2|^2(\xi) + |\widehat{\mathbf{t}}_3|^2(\xi), \xi \in \mathbb{R}^{M \times N}$ and 0 with multiplicity $2MN$ and we can conduct the same analysis as before (see Appendix E). To evaluate if a set of $N_{\mathrm{samples}}$ sampled images is close to the ADSN distribution $p_{\mathrm{data}}$, we evaluate a problem-specific empirical Wasserstein distance: Supposing that the $N_{\mathrm{samples}}$ are drawn from a Gaussian distribution $p^{\mathrm{emp.}} = \mathcal{N}(\mathbf{0}, \boldsymbol{\Gamma})$ such that $\boldsymbol{\Gamma}$ admits the same eigenvectors as $\boldsymbol{\Sigma}$, we compute

$$\mathbf{W}_2^{\mathrm{emp.}}(p^{\mathrm{emp.}}, p_{\mathrm{data}}) = \sqrt{\sum_{\xi \in \mathbb{R}^{3M \times N}} \left( \sqrt{\lambda_1^{\xi,\mathrm{emp.}}} - \sqrt{\lambda_1^{\xi,\mathrm{ADSN}}} \right)^2 + \lambda_2^{\xi,\mathrm{emp.}} + \lambda_3^{\xi,\mathrm{emp.}}} \qquad (26)$$

where $(\lambda_i^{\xi,\mathrm{emp.}})_{\xi \in \mathbb{R}^{M \times N}, 1 \leq i \leq 3}$ are estimators of the eigenvalues of $\boldsymbol{\Gamma}$ given in Appendix F.3.

**Learning the score function.** We train the network using the code[2] associated with the paper Song et al. (2021b). We choose the architecture of DDPM, which is a U-Net described in Ho et al. (2020), with the parameters proposed for the dataset CelebaHQ256 to deal with the $256 \times 256$ ADSN model associated with the top-left image of Figure 3. We use the training procedure corresponding to DDPM cont. in Song et al. (2021b). $\beta$ is linear from 0.05 to 10 with $T = 1$. We train over 1.3M iterations, and we generate at each iteration a new batch of ADSN samples. We implement the stochastic EM and derministic Heun schemes replacing the exact score by its learned version with $N = 1000$ steps and a trunction time $\varepsilon = 10^{-3}$. We name $p_\theta^{\mathrm{EM}}$ and $p_\theta^{\mathrm{Heun}}$, the corresponding distributions and present samples in Figure 3. Both distributions accumulate the four error types.

**Evaluation of the score approximation.** It is not possible to compute theoretically the Wasserstein distance between $p_{\mathrm{data}} = \mathrm{ADSN}(\boldsymbol{u})$ and $p_\theta^{\mathrm{EM}}, p_\theta^{\mathrm{Heun}}$ due to the non-linearity of the learned score. To compute an empirical Wasserstein error between it, we use Equation (26). Let us precise that

---

[2]Code available at `https://github.com/yang-song/score_sde_pytorch`

| | Exact score distribution | | | Learned score distribution | |
|---|---|---|---|---|---|
| $p$ | $\mathbf{W}_2(p, p_{\text{data}})\downarrow$ | $\mathbf{W}_2^{\text{emp.}}(p^{\text{emp.}}, p_{\text{data}})\downarrow$ | $\text{FID}(p^{\text{emp.}}, p_{\text{data}}^{\text{emp.}})\downarrow$ | $\mathbf{W}_2^{\text{emp.}}(p_\theta^{\text{emp.}}, p_{\text{data}}^{\text{emp.}})\downarrow$ | $\text{FID}(p_\theta^{\text{emp.}}, p_{\text{data}}^{\text{emp.}})\downarrow$ |
| EM | 5.16 | 5.1630±7E-5 | 0.0891±8E-4 | 15.6 | 1.02 |
| Heun | 3.73 | 3.7323±2E-4 | 0.0447±6E-4 | 56.7 | 19.4 |

Table 3: **Numerical evaluation of the score approximation for a Gaussian microtexture model.** For two schemes, the EM discretization of the backward SDE and Heun's method associated with the flow ODE, the table shows the Wasserstein distance and FID for theoretical and learned distributions. The theoretical $\mathbf{W}_2$ value is computed with explicit formulas, as done in Table 5. The FID and empirical $\mathbf{W}_2$ w.r.t the theoretical distribution are computed on 25 samplings of 50K images while only one sampling of 50K images is drawn for the parametric distributions (to limit computation time).

this approximation underestimates the real Wasserstein distance since it wrongly assumes that the distributions $p_\theta^{\text{EM}}$, $p_\theta^{\text{Heun}}$ are Gaussian with a covariance matrix diagonalizable in the same basis than the covariance matrix $\Sigma$ of $\text{ADSN}(\boldsymbol{u})$. We complete this dedicated empirical measure with the standard FID. These metrics are reported in Table 3 where for theoretical distributions that are fast to sample we add the standard deviations computed on 25 different $50k$-samplings. For this Gaussian distribution, the score approximation is by far the most impactful source of error, which is in accordance with previous works Chen et al. (2023c); De Bortoli et al. (2021). We observe that the stochastic EM sampling is more resilient to score approximation than the deterministic Heun's scheme, resulting in out-of-distribution samples (Figure 3). We may explain this behavior by recalling the results of Proposition 4 that shows that SDE solutions are less sensitive to initialization errors than ODE. Indeed, adding noise at each iteration tends to mitigate the accumulated errors, and score approximation may be considered as some initialization error ocurring at each step.

# 6 DISCUSSION AND LIMITATIONS

The main limitation of our work is that our results are limited to Gaussian distributions. Resorting to diffusion models for sampling Gaussian distributions is not necessary in practice, rather we use Gaussian distributions as a test case family to provide insight on diffusion models.

A natural extension of this work is to compute error types for more complex distributions (e.g multimodal) such as Gaussian mixtures models (GMM). However, generalizing our results for these more complex distributions one faces two main difficulties. First, to the best of our knowledge, we are unable to derive exact solutions to the backward SDE or the flow ODE under GMM assumption, even though the score has a known analytical expression (Zach et al., 2024; 2023; Shah et al., 2023). Another key feature of this study is to evaluate exactly the Wasserstein error by using Equation (19), strongly relying on the Gaussian assumption. A closed-form of the Wasserstein distance between two GMMs is not known, leading to alternative distance definitions for such models (Delon & Desolneux, 2020). Hence, to compare the distributions generated in practice with exact solutions of time continuous equations under GMM assumption, as we do for the Gaussian case, one should solve two open theoretical problems.

# 7 CONCLUSION

By restricting the analysis of diffusion models to the specific case of Gaussian distributions, we were able to derive exact solutions for both the backward SDE and its associated probability flow ODE. We demonstrate that regarding the initialization error, the SDE sampler is more resilient than the ODE sampler for Gaussian distributions. Additionally, we characterized the discrete Gaussian processes arising when discretizing these equations. This allowed us to provide exact Wasserstein errors for the initialization error, the discretization error, and the truncation error as well as any of their combinations. This theoretical analysis led to conclude that Heun's scheme is the best method out of the four considered schemes, in accordance with empirical previous work (Karras et al., 2022).

To conclude our work we conducted an empirical analysis with a learned score function using standard architecture which showed that the score approximation error may be the most important one in practice. This suggests that assessing the quality of learned score functions is an important research direction for future work.

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

## A  CHARACTERIZATION OF GAUSSIAN DISTRIBUTIONS THROUGH DIFFUSION MODELS

The following proposition shows that our Gaussian assumption occurs if and only if the score function is linear.

**Proposition 5.** *The three following propositions are equivalent:*

*(i)*  $\boldsymbol{x}_0 \sim \mathcal{N}(\boldsymbol{0}, \boldsymbol{\Sigma})$ *for some covariance* $\boldsymbol{\Sigma}$*.*

*(ii)*  $\forall t > 0, \nabla_x \log p_t(\boldsymbol{x})$ *is linear w.r.t* $\boldsymbol{x}$*.*

*(iii)*  $\exists t > 0, \nabla_x \log p_t(\boldsymbol{x})$ *is linear w.r.t* $\boldsymbol{x}$*.*

*In this case, for* $t > 0$*,* $\nabla_{\boldsymbol{x}} \log p_t(\boldsymbol{x}) = -\boldsymbol{\Sigma}_t^{-1}\boldsymbol{x}$*, with* $\boldsymbol{\Sigma}_t$ *defined in Proposition 1.*

*Proof.* $(ii) \Rightarrow (iii)$ is clear.

If $(i)$, for $t > 0$, $p_t(\boldsymbol{x}) = C_t \exp\left(-\frac{1}{2}\boldsymbol{x}^T\boldsymbol{\Sigma}_t^{-1}\boldsymbol{x}\right)$. Consequently, $\nabla_{\boldsymbol{x}} \log p_t(\boldsymbol{x}) = -\boldsymbol{\Sigma}_t^{-1}\boldsymbol{x}$ and $(i) \Rightarrow (ii)$

If $(iii)$, there exists $A$ such that $\nabla_{\boldsymbol{x}} \log p_t(\boldsymbol{x}) = A\boldsymbol{x}$. Consequently, $p_t(\boldsymbol{x}) = C_t \exp(-\frac{1}{2}\boldsymbol{x}^T A\boldsymbol{x})$ and $\boldsymbol{x}_t$ is Gaussian. This provides that $\boldsymbol{x}_0 = e^{B_t}\boldsymbol{x}_t - \boldsymbol{\eta}_t$ is Gaussian and $(iii) \Rightarrow (i)$. $\qquad\square$

## B  PROOFS OF SECTION 3

### B.1  PROPOSITION 1: SOLUTION OF THE FORWARD SDE

We aim at solving:

$$d\boldsymbol{x}_t = -\beta_t\boldsymbol{x}_t dt + \sqrt{2\beta_t}d\boldsymbol{w}_t, \quad \boldsymbol{x}_0 \sim p_{\text{data}}. \tag{27}$$

By considering $\boldsymbol{z}_t = e^{B_t}\boldsymbol{x}_t$ where $B_t = \int_0^t \beta_s ds$,

$$d\boldsymbol{z}_t = \beta_t e^{B_t}\boldsymbol{x}_t + e^{B_t}d\boldsymbol{x}_t = \beta_t e^{B_t}\boldsymbol{x}_t + e^{B_t}(-\beta_t\boldsymbol{x}_t dt + \sqrt{2\beta_t}d\boldsymbol{w}_t) = \sqrt{2\beta_t}e^{B_t}d\boldsymbol{w}_t. \tag{28}$$

Consequently, for $0 \le t \le T$,

$$\boldsymbol{z}_t = \boldsymbol{z}_0 + \int_0^t \sqrt{2\beta_s}e^{B_s}d\boldsymbol{w}_s, \boldsymbol{z}_0 = e^{B_0}\boldsymbol{x}_0 = \boldsymbol{x}_0 \tag{29}$$

and for $0 \le t \le T$,

$$\boldsymbol{x}_t = e^{-B_t}\boldsymbol{z}_t = e^{-B_t}\boldsymbol{x}_0 + e^{-B_t}\int_0^t e^{B_s}\sqrt{2\beta_s}d\boldsymbol{w}_s = e^{-B_t}\boldsymbol{x}_0 + \boldsymbol{\eta}_t. \tag{30}$$

By Itô's isometry (see e.g Øksendal (2010)),

$$\text{Var}\left(\int_0^t e^{B_s}\sqrt{2\beta_s}d\boldsymbol{w}_s\right) = \int_0^t 2\beta_s e^{2B_s}ds = [e^{2B_s}]_0^t = e^{2B_t} - e^{2B_0} = e^{2B_t} - 1 \tag{31}$$

which provides the covariance matrix of $\boldsymbol{\eta}_t$:

$$\text{Cov}\left(\boldsymbol{\eta}_t\right) = e^{-2B_t}(e^{2B_t} - 1)\boldsymbol{I} = \left(1 - e^{-2B_t}\right)\boldsymbol{I}. \tag{32}$$

Because $\boldsymbol{x}_0$ and $\boldsymbol{\eta}_t$ are independent, $\boldsymbol{\Sigma}_t = e^{-2B_t}\boldsymbol{\Sigma} + \left(1 - e^{-2B_t}\right)\boldsymbol{I}$.

And,

$$d\boldsymbol{\Sigma}_t = -2\beta_t e^{-2B_t}(\boldsymbol{\Sigma} - \boldsymbol{I})dt = -2\beta_t\left[\boldsymbol{\Sigma}_t - \boldsymbol{I}\right]dt \tag{33}$$

### B.2 PROPOSITION 2: SOLUTION OF THE BACKWARD SDE UNDER GAUSSIAN ASSUMPTION

We aim at solving

$$dy_t = \beta_{T-t}(y_t - 2\Sigma_{T-t}^{-1}y_t)dt + \sqrt{2\beta_{T-t}}dw_t, \quad 0 \le t \le T \tag{34}$$

Denoting $C_t = \int_0^t \beta_{T-s}ds$, by considering $z_t = \Sigma_{T-t}^{-1}e^{C_t}y_t$,

$$dz_t = e^{C_t}\Sigma_{T-t}^{-1}dy_t + e^{C_t}d[\Sigma_{T-t}^{-1}]y_t + \beta_{T-t}z_tdt \tag{35}$$

$$= e^{C_t}\Sigma_{T-t}^{-1}dy_t - e^{C_t}\Sigma_{T-t}^{-1}d[\Sigma_{T-t}]\Sigma_{T-t}^{-1}y_t + \beta_{T-t}z_tdt \text{ by derivative of the inverse matrix} \tag{36}$$

$$= e^{C_t}\Sigma_{T-t}^{-1}\left[\beta_{T-t}(y_t - 2\Sigma_{T-t}^{-1}y_t)dt + \sqrt{2\beta_{T-t}}dw_t\right] - 2\beta_{T-t}e^{C_t}\Sigma_{T-t}^{-1}\left[\Sigma_{T-t} - I\right]\Sigma_{T-t}^{-1}y_tdt + \beta_{T-t}z_tdt \tag{37}$$

$$\text{(using Equation (9))} \tag{38}$$

$$= \left[\Sigma_{T-t}^{-1}e^{C_t}\beta_{T-t}(y_t - 2\Sigma_{T-t}^{-1}y_t) - \beta_{T-t}z_t + 2\beta_{T-t}\Sigma_{T-t}^{-1}z_t\right]dt + \sqrt{2\beta_{T-t}}e^{C_t}\Sigma_{T-t}^{-1}dw_t \tag{39}$$

$$= \beta_{T-t}(I - 2\Sigma_{T-t}^{-1})z_tdt - \beta_{T-t}z_tdt + 2\beta_{T-t}\Sigma_{T-t}^{-1}z_tdt + e^{C_t}\sqrt{2\beta_{T-t}}\Sigma_{T-t}^{-1}dw_t \tag{40}$$

$$= \sqrt{2\beta_{T-t}}e^{C_t}\Sigma_{T-t}^{-1}dw_t. \tag{41}$$

$$\tag{42}$$

Consequently,

$$z_t = z_0 + \int_0^t \sqrt{2\beta_{T-s}}e^{C_s}\Sigma_{T-s}^{-1}dw_s = \Sigma_T^{-1}y_0 + \int_0^t \sqrt{2\beta_{T-s}}e^{C_s}\Sigma_{T-s}^{-1}dw_s. \tag{43}$$

And,

$$y_t = e^{-C_t}\Sigma_{T-t}z_t = e^{-C_t}\Sigma_{T-t}\Sigma_T^{-1}y_0 + e^{-C_t}\Sigma_{T-t}\int_0^t \Sigma_{T-s}^{-1}e^{C_s}\sqrt{2\beta_{T-s}}dw_s. \tag{44}$$

Finally,

$$y_t = e^{-C_t}\Sigma_{T-t}\Sigma_T^{-1}y_0 + \boldsymbol{\xi}_t \quad \text{with} \quad \boldsymbol{\xi}_t = e^{-C_t}\Sigma_{T-t}\int_0^t \Sigma_{T-s}^{-1}e^{C_s}\sqrt{2\beta_{T-s}}dw_s. \tag{45}$$

By the multidimensional Itô's isometry,

$$\text{Cov}(\int_0^t \Sigma_{T-s}^{-1}e^{C_s}\sqrt{2\beta_{T-s}}dw_s) = 2\int_0^t e^{2C_s}\beta_{T-s}\Sigma_{T-s}^{-2}ds. \tag{46}$$

Now, remark that for $A_s = e^{2C_s}\Sigma_{T-s}^{-1}$,

$$dA_s = 2\beta_{T-s}A_sds + e^{2C_s}d\left[\Sigma_{T-s}^{-1}\right] \tag{47}$$

$$= 2\beta_{T-s}A_sds - 2\beta_{T-s}e^{2C_s}\left[I - \Sigma_{T-s}^{-1}\right]\Sigma_{T-s}^{-1}ds \text{ (using Equation (9))} \tag{48}$$

$$= 2e^{2C_s}\beta_{T-s}\Sigma_{T-s}^{-2}ds. \tag{49}$$

$$\text{Cov}\left(\int_0^t \Sigma_{T-s}^{-1}e^{C_s}\sqrt{\beta_{T-s}}dw_s\right) = \int_0^t dA_s = [A_s]_0^t = e^{2C_t}\Sigma_{T-t}^{-1} - \Sigma_T^{-1}. \tag{50}$$

Finally, $\text{Cov}(\boldsymbol{\xi}_t) = \Sigma_{T-t}^2\left(\Sigma_{T-t}^{-1} - e^{-2C_t}\Sigma_T^{-1}\right) = \Sigma_{T-t} - e^{-2C_t}\Sigma_{T-t}^2\Sigma_T^{-1}$

We have the final formula considering:

$$C_t = \int_0^t \beta_{T-s}ds = \int_{T-t}^T \beta_x dx = \int_0^T \beta_x dx - \int_0^{T-t} \beta_x dx = B_T - B_{T-t} \qquad (51)$$

that provides

$$\mathrm{Cov}(\boldsymbol{y}_t) = \boldsymbol{\Sigma}_{T-t} + e^{-2(B_T - B_{T-t})}\boldsymbol{\Sigma}_{T-t}^2\boldsymbol{\Sigma}_T^{-1}\left(\boldsymbol{\Sigma}_{T-t}^{-1}\mathrm{Cov}(\boldsymbol{y}_0)\boldsymbol{\Sigma}_T^{-1}\boldsymbol{\Sigma}_{T-t} - \boldsymbol{I}\right). \qquad (52)$$

In particular, if $\mathrm{Cov}(\boldsymbol{y}_0)$ and $\boldsymbol{\Sigma}$ commute,

$$\mathrm{Cov}(\boldsymbol{y}_t) = \boldsymbol{\Sigma}_{T-t} + e^{-2(B_T - B_{T-t})}\boldsymbol{\Sigma}_{T-t}^2\boldsymbol{\Sigma}_T^{-1}\left(\boldsymbol{\Sigma}_T^{-1}\mathrm{Cov}(\boldsymbol{y}_0) - \boldsymbol{I}\right). \qquad (53)$$

### B.3 PROPOSITION 3: SOLUTION OF THE ODE PROBABILITY FLOW UNDER GAUSSIAN ASSUMPTION

As done in Khrulkov et al. (2023), the matrix $\boldsymbol{\Sigma}_t^{1/2}$ admits a derivative which is $d\left[\boldsymbol{\Sigma}_t^{1/2}\right] = \frac{1}{2}d\boldsymbol{\Sigma}_t\boldsymbol{\Sigma}_t^{-1/2}$ because it is diagonalisable. Let us check that

$$\boldsymbol{y}_t = \boldsymbol{\Sigma}_T^{-1/2}\boldsymbol{\Sigma}_{T-t}^{1/2}\boldsymbol{y}_0 \qquad (54)$$

is solution of the ODE of Equation (5):

$$d\boldsymbol{y}_t = -\boldsymbol{\Sigma}_T^{-1/2}\frac{1}{2}d\boldsymbol{\Sigma}_{T-t}\boldsymbol{\Sigma}_{T-t}^{-1/2}\boldsymbol{y}_0 \qquad (55)$$

$$= \boldsymbol{\Sigma}_T^{-1/2}\left[\beta_{T-t}\boldsymbol{\Sigma}_{T-t} - \beta_{T-t}\boldsymbol{I}\right]\boldsymbol{\Sigma}_{T-t}^{-1/2}\boldsymbol{y}_0 dt \quad \text{(using Equation (9))} \qquad (56)$$

$$= \left[\beta_{T-t}\boldsymbol{\Sigma}_{T-t} - \beta_{T-t}\boldsymbol{I}\right]\boldsymbol{\Sigma}_{T-t}^{-1}\boldsymbol{\Sigma}_T^{-1/2}\boldsymbol{\Sigma}_{T-t}^{1/2}\boldsymbol{y}_0 dt \quad \text{(by commutativity)} \qquad (57)$$

$$= \left[\beta_{T-t} - \beta_{T-t}\boldsymbol{\Sigma}_{T-t}^{-1}\right]\boldsymbol{y}_t dt \qquad (58)$$

$$= \left[\beta_{T-t} + \beta_{T-t}\nabla_{\boldsymbol{y}}\log p_{T-t}(\boldsymbol{y}_t)\right]\boldsymbol{y}_t dt. \qquad (59)$$

Let us discuss the link between this solution and OT. The formula of OT map between two centered Gaussian distributions $\mathcal{N}(\boldsymbol{0}, \boldsymbol{\Sigma}_1)$ and $\mathcal{N}(\boldsymbol{0}, \boldsymbol{\Sigma}_2)$ is well known. In Peyré & Cuturi (2019), the authors give the linear map (affine when the distributions are not centered) $\boldsymbol{T} : \boldsymbol{X} \mapsto \boldsymbol{A}\boldsymbol{X}$ with

$$\boldsymbol{A} = \boldsymbol{\Sigma}_1^{-\frac{1}{2}}\left(\boldsymbol{\Sigma}_1^{\frac{1}{2}}\boldsymbol{\Sigma}_2\boldsymbol{\Sigma}_1^{\frac{1}{2}}\right)^{\frac{1}{2}}\boldsymbol{\Sigma}_1^{-\frac{1}{2}}. \qquad (60)$$

When $\boldsymbol{\Sigma}_1$ and $\boldsymbol{\Sigma}_2$ commute, this expression simplifies to:

$$\boldsymbol{A} = \boldsymbol{\Sigma}_1^{-1/2}\boldsymbol{\Sigma}_2^{1/2}. \qquad (61)$$

We showed that the solution (Equation (54)) of the backward probability flow in the finite interval $[0, t]$, with $0 \le t \le T$, corresponds to applying to the initial point $\boldsymbol{y}_0$ the linear map

$$\boldsymbol{A} = \boldsymbol{\Sigma}_T^{-\frac{1}{2}}\boldsymbol{\Sigma}_{T-t}^{\frac{1}{2}}, \qquad (62)$$

that is, the OT map between $p_T = \mathcal{N}(\boldsymbol{0}, \boldsymbol{\Sigma}_T)$ and $p_{T-t} = \mathcal{N}(\boldsymbol{0}, \boldsymbol{\Sigma}_{T-t})$.

Let us now derive the covariance matrix of the solution, which characterises a Gaussian distribution.

$$\mathrm{Cov}(\boldsymbol{y}_t) = \boldsymbol{\Sigma}_T^{-1/2}\boldsymbol{\Sigma}_{T-t}^{1/2}\mathrm{Cov}(\boldsymbol{y}_0)\boldsymbol{\Sigma}_{T-t}^{1/2}\boldsymbol{\Sigma}_T^{-1/2}. \qquad (63)$$

In particular, if $\mathrm{Cov}(\boldsymbol{y}_0)$ and $\boldsymbol{\Sigma}$ commute,

$$\mathrm{Cov}(\boldsymbol{y}_t) = \boldsymbol{\Sigma}_T^{-1}\boldsymbol{\Sigma}_{T-t}\mathrm{Cov}(\boldsymbol{y}_0). \qquad (64)$$

### B.4 PROOF OF PROPOSITION 4

For $0 \leq t \leq T$, denoting $(\lambda_{i,t})_{1 \leq i \leq d}$ the eigenvalues of $\mathbf{\Sigma}_t$, the eigenvalues of $\tilde{\mathbf{\Sigma}}_t = \mathrm{Cov}(\tilde{\mathbf{y}}_{T-t})$ are

$$\tilde{\lambda}_{i,t} = \lambda_{i,t} + e^{-2(B_T - B_t)} \lambda_{i,t}^2 \frac{1}{\lambda_{i,T}} \left( \frac{1}{\lambda_{i,T}} - 1 \right), \quad i = 1, \ldots, d. \tag{65}$$

and the eigenvalues of $\widehat{\mathbf{\Sigma}}_t = \mathrm{Cov}(\widehat{\mathbf{y}}_{T-t})$ are

$$\widehat{\lambda}_{i,t} = \frac{\lambda_{i,t}}{\lambda_{i,T}}, \quad 1 \leq i \leq d. \tag{66}$$

Consequently, $\mathbf{W}_2(p_t, \tilde{p}_t)$ is the sum of the squares of all:

$$\sqrt{\lambda_{i,t}} - \sqrt{\tilde{\lambda}_{i,t}} = \sqrt{\lambda_{i,t}} \left( 1 - \sqrt{1 + e^{-2(B_T - B_t)} \lambda_{i,t} \frac{1}{\lambda_{i,T}} \left( \frac{1}{\lambda_{i,T}} - 1 \right)} \right). \tag{67}$$

Similarly, $\mathbf{W}_2(p_t, \widehat{p}_t)$ is the sum of the squares of all:

$$\sqrt{\lambda_{i,t}} - \sqrt{\widehat{\lambda}_{i,t}} = \sqrt{\lambda_{i,t}} \left( 1 - \sqrt{\frac{1}{\lambda_{i,T}}} \right) \tag{68}$$

$$= \sqrt{\lambda_{i,t}} \left( 1 - \sqrt{1 + \left( \frac{1}{\lambda_{i,T}} - 1 \right)} \right). \tag{69}$$

Let us now compare individually these differences.

$$\frac{e^{-2(B_T - B_t)} \lambda_{i,t} \frac{1}{\lambda_{i,T}} \left( \frac{1}{\lambda_{i,T}} - 1 \right)}{\frac{1}{\lambda_{i,T}} - 1} = e^{-2(B_T - B_t)} \frac{\lambda_{i,t}}{\lambda_{i,T}} \tag{70}$$

$$= e^{-2(B_T - B_t)} \frac{e^{-2B_t}(\lambda_i - 1) + 1}{e^{-2B_T}(\lambda_i - 1) + 1} \tag{71}$$

$$= \frac{(\lambda_i - 1) + e^{2B_t}}{(\lambda_i - 1) + e^{2B_T}} \tag{72}$$

$$< 1. \tag{73}$$

**Case 1:** $0 < \lambda_i < 1$ **and** $t > 0$

In this case, $\lambda_{i,T} < 1$ and:

$$0 < e^{-2(B_T - B_t)} \lambda_{i,t} \frac{1}{\lambda_{i,T}} \left( \frac{1}{\lambda_{i,T}} - 1 \right) < \frac{1}{\lambda_{i,T}} - 1. \tag{74}$$

Thus,

$$\left| \sqrt{\lambda_{i,t}} - \sqrt{\tilde{\lambda}_{i,t}} \right| = \sqrt{\tilde{\lambda}_{i,t}} - \sqrt{\lambda_{i,t}} \tag{75}$$

$$= \sqrt{\lambda_{i,t}} \left( \sqrt{1 + e^{-2(B_T - B_t)} \lambda_i^t \frac{1}{\lambda_{i,T}} \left( \frac{1}{\lambda_{i,T}} - 1 \right)} - 1 \right) \tag{76}$$

$$< \sqrt{\lambda_{i,t}} \left( \sqrt{1 + \left( \frac{1}{\lambda_{i,T}} - 1 \right)} - 1 \right) \tag{77}$$

$$= \sqrt{\widehat{\lambda}_{i,t}} - \sqrt{\lambda_{i,t}} \tag{78}$$

$$= \left| \sqrt{\lambda_{i,t}} - \sqrt{\widehat{\lambda}_{i,t}} \right|. \tag{79}$$

**Case 2:** $\lambda_i = 0$ **and** $t = 0$.

In this case, for $1 \leq i \leq d$, $\widehat{\lambda}_{i,T} = \tilde{\lambda}_{i,T} = 0$.

**Case 3:** $\lambda_i = 1$.

In this case, for $1 \leq i \leq d$, $\widehat{\lambda}_{i,t} = \tilde{\lambda}_{i,t} = 1$.

**Case 4:** $1 < \lambda_i$.

In this case, $\lambda_{i,T} \geq 1$, and $\dfrac{e^{-2(B_T - B_t)} \lambda_{i,t} \frac{1}{\lambda_{i,T}} \left( \frac{1}{\lambda_{i,T}} - 1 \right)}{\frac{1}{\lambda_{i,T}} - 1} = e^{-2(B_T - B_t) \frac{\lambda_{i,t}}{\lambda_{i,T}}} < 1$ provides

$$e^{-2(B_T - B_t)} \lambda_{i,t} \frac{1}{\lambda_{i,T}} \left( \frac{1}{\lambda_{i,T}} - 1 \right) > \frac{1}{\lambda_{i,T}} - 1. \tag{80}$$

Finally,

$$\left| \sqrt{\lambda_{i,t}} - \sqrt{\tilde{\lambda}_{i,t}} \right| = \sqrt{\lambda_{i,t}} - \sqrt{\tilde{\lambda}_{i,t}} \tag{81}$$

$$= \sqrt{\lambda_{i,t}} \left( 1 - \sqrt{1 + e^{-2(B_T - B_t)} \lambda_{i,T} \frac{1}{\lambda_{i,T}} \left( \frac{1}{\lambda_{i,T}} - 1 \right)} \right) \tag{82}$$

$$< \sqrt{\lambda_{i,t}} \left( 1 - \sqrt{1 + \left( \frac{1}{\lambda_{i,T}} - 1 \right)} \right) \tag{83}$$

$$= \sqrt{\lambda_{i,t}} - \sqrt{\widehat{\lambda}_{i,t}} \tag{84}$$

$$= \left| \sqrt{\lambda_{i,t}} - \sqrt{\widehat{\lambda}_{i,t}} \right|. \tag{85}$$

This case study provides:

$$\mathbf{W}_2(\tilde{p}_t, p_t) \leq \mathbf{W}_2(\widehat{p}_t, p_t). \tag{86}$$

## C GAUSSIAN CIFAR-10 SAMPLES

The Gaussian CIFAR-10 produces unstructured images. A grid of samples is presented in Figure 4. To sample from this Gaussian, the empirical covariance matrix of size $\mathbb{R}^{(3 \times 32 \times 32) \times (3 \times 32 \times 32)}$ is computed and then the SVD decomposition to extract a square root matrix (see source code).

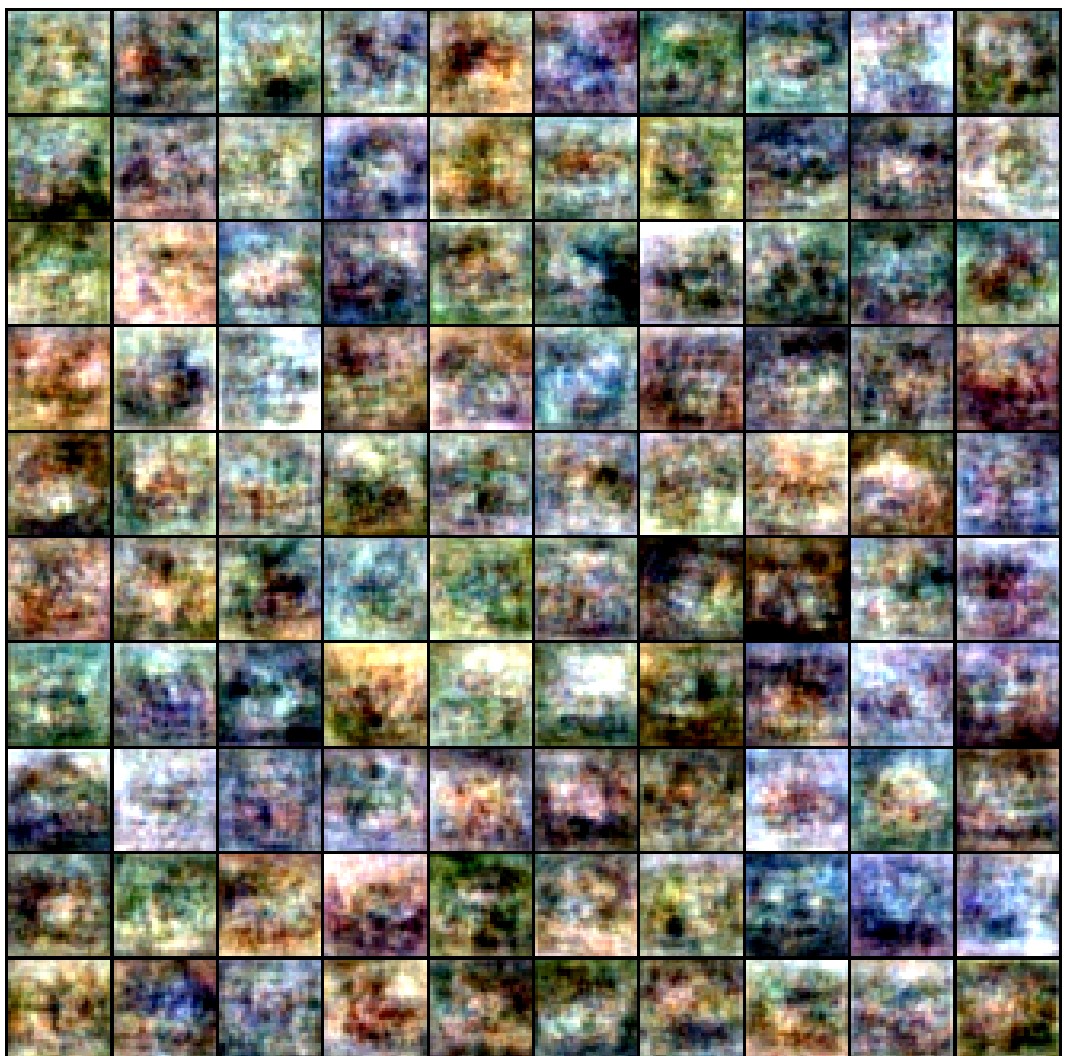

Figure 4: **CIFAR-10 Gaussian samples**. Samples are generated from the Gaussian distribution fitting the CIFAR-10 dataset.

## D   COMPUTATION OF THE 2-WASSERSTEIN DISTANCES FOR NUMERICAL SCHEMES

The 2-Wasserstein errors can be computed by using Equation (19) recalled here:

$$\mathbf{W}_2(\mathcal{N}(\mathbf{0}, \mathbf{\Sigma}_1), \mathcal{N}(\mathbf{0}, \mathbf{\Sigma}_2))^2 = \sum_{1 \leq i \leq d} (\sqrt{\lambda_{i,1}} - \sqrt{\lambda_{i,2}})^2. \tag{19}$$

for two centered Gaussians $\mathcal{N}(\mathbf{0}, \mathbf{\Sigma}_1)$ and $\mathcal{N}(\mathbf{0}, \mathbf{\Sigma}_2)$ such that $\mathbf{\Sigma}_1, \mathbf{\Sigma}_2$ are simultaneously diagonalizable with respective eigenvalues $(\lambda_{i,1})_{1 \leq i \leq d}, (\lambda_{i,2})_{1 \leq i \leq d}$. We aim at computing these errors between the Gaussian process following $(p_t)_{0 \leq t \leq T}$ and the processes induced by the discretization schemes. Table 1 shows that each discretization scheme leads to a discrete time Gaussian process whose covariance matrix is diagonalizable in the basis of $\mathbf{\Sigma}$ when intialize them with either $\mathcal{N}_0$ (usual sampling) or $p_T$ (no initialization error). Let detail this point for the Euler-Maruyama (EM) scheme. Let denote $(\boldsymbol{v}_i)_{1 \leq i \leq d}$ a basis of eigenvectors of $\mathbf{\Sigma}$ and its associated eigenvalues $(\lambda_i)_{1 \leq i \leq d}$. For

$0 \leq t \leq T$, $\mathbf{\Sigma}_t = e^{-2B_t}\mathbf{\Sigma} + (1 - e^{-2B_t})\boldsymbol{I}$ and for all $1 \leq i \leq d$,

$$\mathbf{\Sigma}_t \boldsymbol{v}_i = \left(e^{-2B_t}\lambda_i + (1 - e^{-2B_t})\right)\boldsymbol{v}_i \tag{87}$$

Consequently $\mathbf{\Sigma}_t$ admits the same eigenvectors than $\mathbf{\Sigma}$ with associated eigenvalues $(\lambda_{i,t})_{1 \leq i \leq d} = \left(e^{-2B_t}\lambda_i + (1 - e^{-2B_t})\right)_{1 \leq i \leq d}$. Let study the covariance matrix of the EM process. Let denote $\left(\mathbf{\Sigma}_k^{\Delta,\text{EM}}\right)_{1 \leq i \leq d, 0 \leq k \leq N}$ the covariance matrix of the Gaussian process generated by the EM scheme at each step and $\left(\lambda_{i,k}^{\Delta,\text{EM}}\right)_{1 \leq i \leq d, 0 \leq k \leq N}$ its eigenvalues. First,

$$\mathbf{\Sigma}_0^{\Delta,\text{EM}} = \begin{cases} \boldsymbol{I} & \text{if } \boldsymbol{y}_T \text{ is initialized at } \mathcal{N}_0 \\ \mathbf{\Sigma}_T & \text{if } \boldsymbol{y}_T \text{ is initialized at } p_T \end{cases}. \tag{88}$$

And consequently,

$$\lambda_{i,0}^{\Delta,\text{EM}} = \begin{cases} 1 & \text{if } \boldsymbol{y}_T \text{ is initialized at } \mathcal{N}_0 \\ e^{-2B_T}\lambda_i + (1 - e^{-2B_T}) & \text{if } \boldsymbol{y}_T \text{ is initialized at } p_T \end{cases} \quad 1 \leq i \leq d. \tag{89}$$

Then, by Table 1,

$$\tilde{\boldsymbol{y}}_1^{\Delta,\text{EM}} = \left(\boldsymbol{I} + \Delta_t\beta_{T-t_0}\left(\boldsymbol{I} - 2\mathbf{\Sigma}_{T-t_0}^{-1}\right)\right)\tilde{\boldsymbol{y}}_0^{\Delta,\text{EM}} + \sqrt{2\Delta_t\beta_{T-t_0}}\boldsymbol{z}_0, \ \boldsymbol{z}_0 \sim \mathcal{N}_0 \tag{90}$$

and

$$\mathbf{\Sigma}_1^{\Delta,\text{EM}} = \left(\boldsymbol{I} + \Delta_t\beta_{T-t_0}\left(\boldsymbol{I} - 2\mathbf{\Sigma}_{T-t_0}^{-1}\right)\right)\mathbf{\Sigma}_0^{\Delta,\text{EM}}\left(\boldsymbol{I} + \Delta_t\beta_{T-t_0}\left(\boldsymbol{I} - 2\mathbf{\Sigma}_{T-t_0}^{-1}\right)\right)^T + 2\Delta_t\beta_{T-t_0}\boldsymbol{I} \tag{91}$$

$$= \left(\boldsymbol{I} + \Delta_t\beta_{T-t_0}\left(\boldsymbol{I} - 2\mathbf{\Sigma}_{T-t_0}^{-1}\right)\right)^2\mathbf{\Sigma}_0^{\Delta,\text{EM}} + 2\Delta_t\beta_{T-t_0}\boldsymbol{I} \text{ because } \mathbf{\Sigma} \text{ and } \mathbf{\Sigma}_0^{\Delta,\text{EM}} \text{ commute.} \tag{92}$$

Let $1 \leq i \leq d$,

$$\mathbf{\Sigma}_1^{\Delta,\text{EM}}\boldsymbol{v}_i = \left[\left(1 + \Delta_t\beta_{T-t_0}\left(\boldsymbol{I} - \frac{2}{\lambda_{i,T-t_0}}\right)\right)^2\lambda_{i,0}^{\Delta,\text{EM}} + 2\Delta_t\beta_{T-t_0}\right]\boldsymbol{v}_i \tag{93}$$

Consequently, $(\boldsymbol{v}_i)_{1 \leq i \leq d}$ is also a basis of eigenvectors of $\mathbf{\Sigma}_1^{\Delta,\text{EM}}$ and

$$\lambda_{i,1}^{\Delta,\text{EM}} = \left(1 + \Delta_t\beta_{T-t_0}\left(\boldsymbol{I} - \frac{2}{\lambda_{i,T-t_0}}\right)\right)^2\lambda_{i,0}^{\Delta,\text{EM}} + 2\Delta_t\beta_{T-t_0}, \quad 1 \leq i \leq d. \tag{94}$$

Thus, we can obtain the eigenvalues $\left(\lambda_{i,k}^{\Delta,\text{EM}}\right)_{1 \leq i \leq d, 0 \leq k \leq N}$ at each time and plot at each time

$$\sqrt{\sum_{1 \leq i \leq d}\left(\sqrt{\lambda_{i,T-t_k}} - \sqrt{\lambda_{i,k}^{\Delta,\text{EM}}}\right)}, \quad 1 \leq k \leq N \tag{95}$$

as done in Figure 1. These computations can be led for the different schemes, as presented in Table 4.

## E  THEORETICAL WASSERSTEIN DISTANCE FOR THE ADSN MODEL

As done for the Gaussian CIFAR-10, the Wasserstein errors can be computed for the ADSN model as shown in Figure 5 and Table 5.

## F  STUDY OF THE COVARIANCE MATRIX OF THE ADSN DISTRIBUTION

### F.1  REMINDERS ON THE DISCRETE FOURIER TRANSFORM (DFT)

For a given image $\boldsymbol{v} \in \mathbb{R}^{3 \times M \times N}$, we define the DFT of $\boldsymbol{v}$, $\widehat{\boldsymbol{v}} \in \mathbb{R}^{3 \times M \times N}$ such that for $1 \leq c \leq 3$, $\xi \in \mathbb{R}^{M \times N}$

| | | |
|---|---|---|
| **SDE schemes** | EM | $\lambda_{i,k+1}^{\Delta,\text{EM}} = \left(1 + \Delta_t \beta_{T-t_k}\left(1 - \frac{2}{\lambda_{i,T-t_k}}\right)\right)^2 \lambda_{i,k}^{\Delta,\text{EM}} + 2\Delta_t \beta_{T-t_k}, \quad 1 \leq i \leq d, 0 \leq k \leq N-1$ |
| | EI | $\lambda_{i,k+1}^{\Delta,\text{EI}} = \left(1 + \gamma_{1,k}\left(1 - \frac{2}{\lambda_{i,T-t_k}}\right)\right)^2 \lambda_{i,k}^{\Delta,\text{EI}} + 2\gamma_{2,k} \quad 1 \leq i \leq d, 0 \leq k \leq N-1$ |
| **ODE schemes** | Euler | $\lambda_{i,k+1}^{\Delta,\text{Euler}} = \left(1 + \Delta_t \beta_{T-t_k}\left(1 - \frac{1}{\lambda_{i,T-t_k}}\right)\right)^2 \lambda_i^{\text{Euler},k} \quad 1 \leq i \leq d, 0 \leq k \leq N-1$ |
| | Heun | $\lambda_{i,k+1}^{\Delta,\text{Heun}} = \left(1 + \frac{\Delta_t}{2}\beta_{T-t_k}\left(1 - \frac{1}{\lambda_{i,T-t_k}}\right) + \frac{\Delta_t}{2}\beta_{T-t_{k+1}}\left(1 - \frac{1}{\lambda_{i,T-t_{k+1}}}\right)\left(1 + \Delta_t \beta_{T-t_k}\left(1 - \frac{1}{\lambda_{i,T-t_k}}\right)\right)\right)\lambda_{i,k}^{\Delta,\text{Heun}}$ $1 \leq i \leq d, 0 \leq k \leq N-1$ |

Table 4: Recursive form of the eigenvalues of the covariance matrix associated with the Gaussian process generated by the different schemes for a regular time schedule $(t_k)_{0 \leq k \leq N}$ with steps $\Delta_t = \frac{T}{N}$.

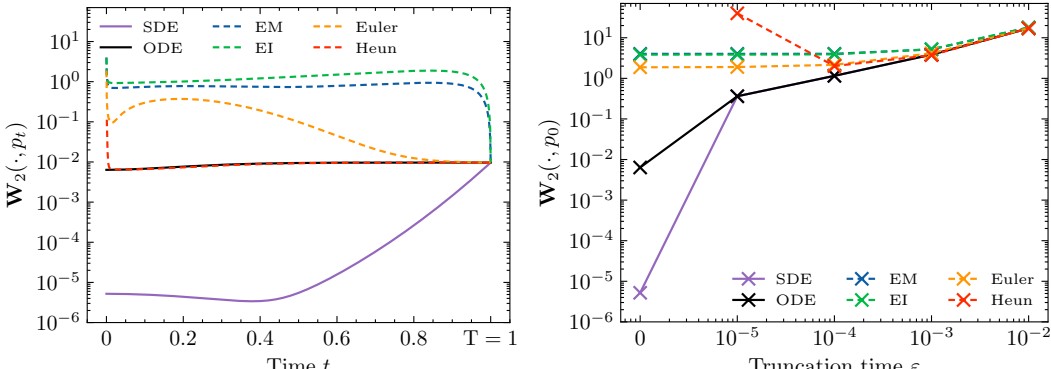

(a) Initialization error along the integration time.  (b) Truncation error for different truncation time $\varepsilon$.

Figure 5: **Wasserstein errors for the diffusion models associated with the Gaussian microtextures.** Left: Evolution of the Wasserstein distance between $p_t$ and the distributions associated with the continuous SDE, the continuous flow ODE and four discrete sampling schemes with standard $\mathcal{N}_0$ initialization, either stochastic (Euler-Maruyama (EM) and Exponential Integrator (EI)) or deterministic (Euler and Heun). While the continuous SDE is less sensible than the continuous ODE (as proved by Proposition 4), the initialization error impacts all discrete schemes. Heun's method has the lowest error and is very close to the theoretical ODE, except for the last step that is usually discarded when using time truncation. Right: Wasserstein errors due to time truncation for various truncation times $\epsilon$. Heun's scheme is not defined without truncation time due to the zero eigenvalue. Interestingly, for the standard practice truncation time $\varepsilon = 10^{-3}$, all numerical schemes have a comparable error close to their continuous counterparts.

$$\widehat{\boldsymbol{v}}_{c,\xi} = \sum_{x \in M \times N} \boldsymbol{v}_{c,x} \exp\left(-\frac{2i\pi x_1 \xi_1}{M}\right) \exp\left(-\frac{2i\pi x_2 \xi_2}{N}\right), \quad i^2 = -1 \tag{96}$$

where $\widehat{\boldsymbol{v}}_{c,\xi}$ is the value of $\widehat{\boldsymbol{v}}$ at coordinate $\xi$ of the k-th channel of $\widehat{\boldsymbol{v}}$. For $\boldsymbol{u} \in \mathbb{R}^{3M \times N}$, by defining $\boldsymbol{u} \star \boldsymbol{v}$ the periodic convolution such that for $1 \leq c \leq 3, x \in \mathbb{R}^{M \times N}$:

$$(\boldsymbol{u} \star \boldsymbol{v})_{c,x} = \sum_{y \in M \times N} \boldsymbol{u}_{c,x-y} \boldsymbol{v}_{c,y} \tag{97}$$

we have:

$$\widehat{\boldsymbol{u} \star \boldsymbol{v}} = \widehat{\boldsymbol{u}} \odot \widehat{\boldsymbol{v}}, \tag{98}$$

where $\odot$ is the componentwise product.

| | | Continuous | | $N=50$ | | $N=250$ | | $N=500$ | | $N=1000$ | |
|---|---|---|---|---|---|---|---|---|---|---|---|
| | | $p_T$ | $\mathcal{N}_0$ | $p_T$ | $\mathcal{N}_0$ | $p_T$ | $\mathcal{N}_0$ | $p_T$ | $\mathcal{N}_0$ | $p_T$ | $\mathcal{N}_0$ |
| EM | $\varepsilon=0$ | 0 | 5.2E-06 | 53.37 | 53.37 | 10.58 | 10.58 | 6.27 | 6.27 | 4.02 | 4.02 |
| | $\varepsilon=10^{-5}$ | 0.36 | 0.36 | 53.35 | 53.35 | 10.57 | 10.57 | 6.26 | 6.26 | 4.02 | 4.02 |
| | $\varepsilon=10^{-4}$ | 1.15 | 1.15 | 53.21 | 53.21 | 10.53 | 10.53 | 6.25 | 6.25 | 4.03 | 4.03 |
| | $\varepsilon=10^{-3}$ | 3.84 | 3.84 | 51.92 | 51.92 | 10.55 | 10.55 | 6.80 | 6.80 | 5.16 | 5.16 |
| EI | $\varepsilon=0$ | 0 | 5.2E-06 | 30.91 | 30.91 | 8.85 | 8.85 | 5.71 | 5.71 | 3.84 | 3.84 |
| | $\varepsilon=10^{-5}$ | 0.36 | 0.36 | 30.92 | 30.92 | 8.85 | 8.85 | 5.72 | 5.72 | 3.84 | 3.84 |
| | $\varepsilon=10^{-4}$ | 1.15 | 1.15 | 31.01 | 31.01 | 8.92 | 8.92 | 5.78 | 5.78 | 3.90 | 3.90 |
| | $\varepsilon=10^{-3}$ | 3.84 | 3.84 | 31.94 | 31.94 | 9.74 | 9.74 | 6.76 | 6.76 | 5.24 | 5.24 |
| Euler | $\varepsilon=0$ | 0 | 6.4E-03 | 5.69 | 5.70 | 3.27 | 3.27 | 2.50 | 2.51 | 1.87 | 1.87 |
| | $\varepsilon=10^{-5}$ | 0.36 | 0.36 | 5.70 | 5.71 | 3.28 | 3.28 | 2.53 | 2.53 | 1.90 | 1.90 |
| | $\varepsilon=10^{-4}$ | 1.15 | 1.15 | 5.80 | 5.80 | 3.43 | 3.43 | 2.72 | 2.72 | 2.15 | 2.15 |
| | $\varepsilon=10^{-3}$ | 3.84 | 3.84 | 6.79 | 6.79 | 4.85 | 4.85 | 4.41 | 4.41 | 4.14 | 4.14 |
| Heun | $\varepsilon=0$ | 0 | 6.4E-03 | - | - | - | - | - | - | - | - |
| | $\varepsilon=10^{-5}$ | 0.36 | 0.36 | 2.4E+03 | 2.4E+03 | 3.0E+02 | 3.0E+02 | 1.1E+02 | 1.1E+02 | 40.00 | 40.00 |
| | $\varepsilon=10^{-4}$ | 1.15 | 1.15 | 2.3E+02 | 2.3E+02 | 26.34 | 26.34 | 8.54 | 8.54 | 2.01 | 2.01 |
| | $\varepsilon=10^{-3}$ | 3.84 | 3.84 | 15.42 | 15.42 | 2.25 | 2.25 | 3.40 | 3.40 | 3.73 | 3.73 |

Table 5: **Ablation study of Wasserstein errors for the Gaussian microtextures.** For a given discretization scheme, the table presents the Wasserstein distance associated with the truncation error for different values of $\varepsilon$. The columns $p_T$ and $\mathcal{N}_0$ show the influence of the initialization error. The continuous column corresponds to the continuous SDE or ODE linked with the scheme (identical values for EM, EI and Euler, Heun). Note that the Heun scheme is not defined without truncation time due to the zero eigenvalue.

### F.2 EIGENVECTORS OF THE COVARIANCE MATRIX OF THE ADSN DISTRIBUTION

Let $\boldsymbol{u} \in \mathbb{R}^{3 \times M \times N}$ and its associated texton $\mathbf{t} \in \mathbb{R}^{3 \times M \times N}$. The distribution $\mathrm{ADSN}(\boldsymbol{u})$ is the Gaussian distribution of $\boldsymbol{X} = \mathbf{t} \star \boldsymbol{w}$ such that:

$$\boldsymbol{X}_i = \mathbf{t}_i \star \boldsymbol{w} \in \mathbb{R}^{M \times N}, 1 \le i \le 3, \boldsymbol{w} \sim \mathcal{N}_0 \tag{99}$$

Consequently, denoting $\boldsymbol{\Sigma}$ the covariance of $\mathrm{ADSN}(\boldsymbol{u})$, for $\boldsymbol{v} \in \mathbb{R}^{3M \times N}$,

$$\widehat{\boldsymbol{\Sigma}\boldsymbol{v}}_i = \widehat{\mathbf{t}}_i\overline{\widehat{\mathbf{t}}_1}\widehat{\boldsymbol{v}}_1 + \widehat{\mathbf{t}}_i\overline{\widehat{\mathbf{t}}_2}\widehat{\boldsymbol{v}}_2 + \widehat{\mathbf{t}}_i\overline{\widehat{\mathbf{t}}_3}\widehat{\boldsymbol{v}}_3 = \widehat{\mathbf{t}}_i\left(\overline{\widehat{\mathbf{t}}_1}\widehat{\boldsymbol{v}}_1 + \overline{\widehat{\mathbf{t}}_2}\widehat{\boldsymbol{v}}_2 + \overline{\widehat{\mathbf{t}}_3}\widehat{\boldsymbol{v}}_3\right) \tag{100}$$

This equation proves that the kernel of $\boldsymbol{\Sigma}$ contains the kernel of $\boldsymbol{v} \in \mathbb{R}^{3 \times M \times N} \mapsto \overline{\widehat{\mathbf{t}}_1}\widehat{\boldsymbol{v}}_1 + \overline{\widehat{\mathbf{t}}_2}\widehat{\boldsymbol{v}}_2 + \overline{\widehat{\mathbf{t}}_3}\widehat{\boldsymbol{v}}_3 \in \mathbb{R}^{M \times N}$ which has a dimension greater than $2MN$. Consequently, $0$ is eigenvalue of $\boldsymbol{\Sigma}$ with multiplicity greater than $2MN$. Furthermore, for $\xi \in \mathbb{R}^{M \times N}$, denoting $\boldsymbol{u}^{1,\xi}$ such that:

$$\widehat{\boldsymbol{u}}_i^{1,\xi}(\omega) = \mathbf{1}_{\omega=\xi}\widehat{\mathbf{t}}_i(\omega), 1 \le i \le 3, \omega \in \mathbb{R}^{M \times N} \tag{101}$$

we have,

$$\boldsymbol{\Sigma}\boldsymbol{u}^{1,\xi} = (|\widehat{\mathbf{t}}_1(\xi)|^2 + |\widehat{\mathbf{t}}_2(\xi)|^2 + |\widehat{\mathbf{t}}_3(\xi)|^2)\boldsymbol{u}^{1,\xi}. \tag{102}$$

Furthermore, the family $\left(\boldsymbol{u}^{1,\xi}\right)_{\xi \in M \times N}$ is orthogonal. Thus, the eigenvalues of $\boldsymbol{\Sigma}$ are $\left(|\widehat{\mathbf{t}}_1(\xi)|^2 + |\widehat{\mathbf{t}}_2(\xi)|^2 + |\widehat{\mathbf{t}}_3(\xi)|^2\right)_{\xi \in M \times N}$ and $0$ with multiplicity $2MN$.

For $\xi \in \mathbb{R}^{M \times N}$, we denote $\boldsymbol{u}^{2,\xi}, \boldsymbol{u}^{3,\xi}$ such that for $\omega \in \mathbb{R}^{M \times N}$:

$$
\begin{cases}
\widehat{\boldsymbol{u}}_1^{2,\xi}(\omega) & = -\mathbf{1}_{\omega=\xi}\overline{\bar{\mathbf{t}}}_3(\omega) \\
\widehat{\boldsymbol{u}}_2^{2,\xi}(\omega) & = 0 \\
\widehat{\boldsymbol{u}}_3^{2,\xi}(\omega) & = \mathbf{1}_{\omega=\xi}\overline{\bar{\mathbf{t}}}_1(\omega)
\end{cases}
\tag{103}
$$

$$
\begin{cases}
\widehat{\boldsymbol{u}}_1^{3,\xi}(\omega) & = 0 \\
\widehat{\boldsymbol{u}}_2^{3,\xi}(\omega) & = -\mathbf{1}_{\omega=\xi}\overline{\bar{\mathbf{t}}}_3(\omega) \\
\widehat{\boldsymbol{u}}_3^{3,\xi}(\omega) & = \mathbf{1}_{\omega=\xi}\overline{\bar{\mathbf{t}}}_2(\omega)
\end{cases}
\tag{104}
$$

We have

$$
\boldsymbol{\Sigma}\boldsymbol{u}^{2,\xi} = 0.\boldsymbol{u}^{2,\xi}
\tag{105}
$$

$$
\boldsymbol{\Sigma}\boldsymbol{u}^{3,\xi} = 0.\boldsymbol{u}^{3,\xi}.
\tag{106}
$$

Then, applying the orthonomalization of Gram-Schmidt on each tuple $(\boldsymbol{u}^{1,\xi},\boldsymbol{u}^{2,\xi},\boldsymbol{u}^{3,\xi})_{\xi\in\mathbb{R}^{M\times N}}$, we obtain an orthonormal basis in the Fourier domain $(\boldsymbol{v}^{1,\xi},\boldsymbol{v}^{2,\xi},\boldsymbol{v}^{3,\xi})_{\xi\in\mathbb{R}^{M\times N}}$ of eigenvectors of $\boldsymbol{\Sigma}$. More precisely, for $\xi_1,\xi_2\in\mathbb{R}^{M\times N}, 1\leq j_1,j_2\leq 3$,

$$
\left(\overline{\widehat{\boldsymbol{v}}}^{j_1,\xi_1}\right)^T\widehat{\boldsymbol{v}}^{j_2,\xi_2} = \sum_{\substack{x_1\in M\times N \\ x_2\in M\times N}} \overline{\widehat{\boldsymbol{v}}}_{x_1}^{j_1,\xi_1}\widehat{\boldsymbol{v}}_{x_2}^{j_2,\xi_2}
\tag{107}
$$

$$
= \mathbf{1}_{\substack{j_1=j_2 \\ \xi_1=\xi_2}}
\tag{108}
$$

which is applying the square root of $\boldsymbol{\Sigma}$ to the white Gaussian noise $\boldsymbol{w}$. Furthermore, we can ensure that for $\xi\neq\omega\in\mathbb{R}^{M\times N}, 1\leq j\leq 3, \widehat{\boldsymbol{v}}^{j,\xi}(\omega)=0$ such that only the frequency $\xi$ is active in the Fourier transform of $\boldsymbol{v}^{j,\xi}$. Consequently, for $\boldsymbol{w}\in\mathbb{R}^{3M\times N}$,

$$
\overline{\widehat{\boldsymbol{w}}}^T\boldsymbol{v}^{j,\xi} = \sum_{1\leq i\leq 3}\overline{\widehat{\boldsymbol{w}}}_i(\xi)\widehat{\boldsymbol{v}}_i^{j,\xi}(\xi).
\tag{109}
$$

In particular,

$$
\left(\overline{\widehat{\boldsymbol{v}}}^{j,\xi}\right)^T = \|\widehat{\boldsymbol{v}}^{j,\xi}\|^2 = \sum_{1\leq i\leq 3}\left|\boldsymbol{v}_i^{j,\xi}(\xi)\right|^2 = 1.
\tag{110}
$$

### F.3 COMPUTATION OF THE EMPIRICAL WASSERSTEIN ERROR IN THE ADSN COVARIANCE DIAGONALIZATION BASIS

Let consider a Gaussian distribution $\mathcal{N}(\mathbf{0},\boldsymbol{\Gamma})$ such that there exists $(\lambda_1^\xi,\lambda_2^\xi,\lambda_3^\xi)_{\xi\in\mathbb{R}^{M\times N}}$ such that for all $\xi\in\mathbb{R}^{M\times N}$,

$$
\boldsymbol{\Gamma}\boldsymbol{v}^{j,\xi} = \lambda_j^\xi\boldsymbol{v}^{j,\xi}, \quad 1\leq j\leq 3.
\tag{111}
$$

Let $\boldsymbol{w}\sim\mathcal{N}_0\in\mathbb{R}^{3M\times N}, (\boldsymbol{v}^{1,\xi},\boldsymbol{v}^{2,\xi},\boldsymbol{v}^{3,\xi})_{\xi\in\mathbb{R}^{M\times N}}$ is an orthonormal basis in the Fourier domain such that:

$$
\widehat{\boldsymbol{w}} = \sum_{\xi\in\mathbb{R}^{M\times N}}\left(\left[\overline{\widehat{\boldsymbol{w}}}^T\widehat{\boldsymbol{v}}^{1,\xi}\right]\widehat{\boldsymbol{v}}^{1,\xi} + \left[\overline{\widehat{\boldsymbol{w}}}^T\widehat{\boldsymbol{v}}^{2,\xi}\right]\widehat{\boldsymbol{v}}^{2,\xi} + \left[\overline{\widehat{\boldsymbol{w}}}^T\widehat{\boldsymbol{v}}^{3,\xi}\right]\widehat{\boldsymbol{v}}^{3,\xi}\right)
\tag{112}
$$

$$
\tag{113}
$$

A sample drawn from $\mathcal{N}(\mathbf{0},\boldsymbol{\Gamma})$ has the same distribution as $\boldsymbol{Y}$ given by

$$\widehat{\boldsymbol{Y}} = \sum_{\xi \in \mathbb{R}^{M \times N}} \sqrt{\lambda_1^\xi} \left[ \overline{\widehat{\boldsymbol{w}}}^T \widehat{\boldsymbol{v}}^{1,\xi} \right] \widehat{\boldsymbol{v}}^{1,\xi} + \sum_{\xi \in \mathbb{R}^{M \times N}} \sqrt{\lambda_2^\xi} \left[ \overline{\widehat{\boldsymbol{w}}}^T \widehat{\boldsymbol{v}}^{2,\xi} \right] \widehat{\boldsymbol{v}}^{2,\xi} + \sum_{\xi \in \mathbb{R}^{M \times N}} \sqrt{\lambda_3^\xi} \left[ \overline{\widehat{\boldsymbol{w}}}^T \widehat{\boldsymbol{v}}^{3,\xi} \right] \widehat{\boldsymbol{v}}^{3,\xi}. \tag{114}$$

Note that the three channels of $\boldsymbol{w}$ are independent. Furthermore, for $1 \le j \le 3$

$$\left( \overline{\widehat{\boldsymbol{v}}}^{j,\xi} \right)^T \widehat{\boldsymbol{Y}} = \sqrt{\lambda_1^\xi} \left[ \overline{\widehat{\boldsymbol{w}}}^T \widehat{\boldsymbol{v}}^{j,\xi} \right] \left\| \widehat{\boldsymbol{v}}^{j,\xi} \right\|^2 = \sqrt{\lambda_1^\xi} \left[ \overline{\widehat{\boldsymbol{w}}}^T \widehat{\boldsymbol{v}}^{j,\xi} \right] \tag{115}$$

$$\left| \left( \overline{\widehat{\boldsymbol{v}}}^{j,\xi} \right)^T \widehat{\boldsymbol{Y}} \right|^2 = \lambda_j^\xi \left| \overline{\widehat{\boldsymbol{w}}}^T \widehat{\boldsymbol{v}}^{j,\xi} \right|^2 \tag{116}$$

$$\mathbb{E} \left[ \left| \left( \overline{\widehat{\boldsymbol{v}}}^{j,\xi} \right)^T \widehat{\boldsymbol{Y}} \right|^2 \right] = \lambda_j^\xi \mathbb{E} \left[ \left| \overline{\widehat{\boldsymbol{w}}}^T \widehat{\boldsymbol{v}}^{j,\xi} \right|^2 \right] \tag{117}$$

$$\mathbb{E} \left[ \left| \overline{\widehat{\boldsymbol{w}}}^T \widehat{\boldsymbol{v}}^{j,\xi} \right|^2 \right] = \sum_{1 \le c_1, c_2 \le 3} \mathbb{E} \left[ \overline{\widehat{\boldsymbol{w}}}_{c_1}(\xi) \widehat{\boldsymbol{w}}_{c_2}(\xi) \right] \widehat{\boldsymbol{v}}_{c_1}^{j,\xi}(\xi) \overline{\widehat{\boldsymbol{v}}}_{c_2}(\xi) \text{ by Equation (109)} \tag{118}$$

$$= \sum_{1 \le c \le 3} \mathbb{E} \left[ |\widehat{\boldsymbol{w}}_c(\xi)|^2 \right] \left| \widehat{\boldsymbol{v}}_c^{j,\xi}(\xi) \right|^2 \text{ because the channels are inependent} \tag{119}$$

$$= 3MN \sum_{1 \le c \le 3} \left| \widehat{\boldsymbol{v}}_c^{j,\xi}(\xi) \right|^2 \text{ because } \mathbb{E} \left[ |\widehat{\boldsymbol{w}}_c(\xi)|^2 \right] = MN \tag{120}$$

$$= 3MN \text{ by Equation (110).} \tag{121}$$

Finally,

$$\mathbb{E} \left[ \left| \left( \overline{\widehat{\boldsymbol{v}}}^{j,\xi} \right)^T \widehat{\boldsymbol{Y}} \right|^2 \right] = 3MN \lambda_1^\xi \tag{122}$$

Finally, for a given sampling $(\boldsymbol{Y}_k)_{1 \le k \le N_{\text{samples}}}$ following the distribution $\mathcal{N}(\boldsymbol{0}, \boldsymbol{\Gamma})$, an estimator of $\lambda_j^\xi$ is:

$$\lambda_j^{\xi,\text{emp.}} = \frac{1}{3N_{\text{samples}} MN} \sum_{k=1}^{N_{\text{samples}}} \left| \left( \overline{\widehat{\boldsymbol{v}}}^{j,\xi} \right)^T \widehat{\boldsymbol{Y}}_k \right|^2. \tag{123}$$

The empirical Wasserstein distance between the Gaussian distribution $\mathcal{N}(\boldsymbol{0}, \Gamma)$ and the ADSN model with texton $\mathbf{t}$ is:

$$\mathbf{W}_2^{\text{emp.}} (\mathcal{N}^{\text{emp.}}(\boldsymbol{0}, \boldsymbol{\Gamma}), \text{ADSN}(\boldsymbol{u})) = \sqrt{\sum_{\xi \in \mathbb{R}^{M \times N}} \left( \left( \sqrt{\lambda_1^{\xi,\text{emp.}}} - \sqrt{\lambda_1^{\xi,\text{ADSN}}} \right)^2 + \lambda_2^{\xi,\text{emp.}} + \lambda_3^{\xi,\text{emp.}} \right)} \tag{124}$$

with $\lambda_1^{\xi,\text{ADSN}} = |\widehat{\mathbf{t}}_1(\xi)|^2 + |\widehat{\mathbf{t}}_2(\xi)|^2 + |\widehat{\mathbf{t}}_3(\xi)|^2$ for $\xi \in \mathbb{R}^{M \times N}$.

Furthermore, the computations can be vectorized by componentwise products in the Fourier domain.