# OpenReview forum: "Diffusion models for Gaussian distributions: Exact solutions and Wasserstein errors"
_ICLR.cc/2025/Conference — Submitted to ICLR 2025_

### Official Review · Reviewer_FEtE · 2024-10-28

**Soundness:** 3
**Presentation:** 3
**Contribution:** 3
**Rating:** 6
**Confidence:** 5

**Summary:**

This paper studies theoretically the behavior of diffusion models and their numerical implementation when the data distribution is Gaussian. In this restricted framework where the score function is a linear operator, we derive the analytical solutions of the backward SDE and the probability flow ODE. We prove that these solutions and their discretizations are all Gaussian processes, which allows us to compute
exact Wasserstein errors induced by each error type for any sampling scheme.

**Strengths:**

1.The paper derives exact analytical solutions for the forward and inverse stochastic differential equations (SDEs), as well as the probabilistic flow ordinary differential equations (ODEs), within the framework of Gaussian distributions.
2.The initialization error, truncation error, discretization error, and  score approximation error  are systematically analyzed.
3.Different numerical sampling schemes are compared through numerical experiments, and in particular, the effectiveness of Heun's method for Gaussian distributions is demonstrated.

**Weaknesses:**

1.While the paper provides an in-depth theoretical analysis, it may lack validation on diverse real-world datasets.
2.Although it is possible to derive exact solutions under the assumption of a Gaussian distribution, obtaining such analytical solutions for more complex data distributions may be challenging, which could limit the generalization of the theory.
3.The paper points out that score approximation errors may be significant in practical applications but does not propose an effective solution for minimizing these errors.

**Questions:**

1.The paper compares several numerical sampling schemes, but has it covered all the optimal possible schemes?
2.For complex data distributions, analytic solutions may be difficult to obtain, which may affect the computational efficiency of the model. Does the paper consider the issue of computational efficiency in practical applications?

---

> ### Author Response · Authors · 2024-11-17
>
> We thank  you for your positive reviews on our paper.
>
>
> > 1.The paper compares several numerical sampling schemes, but has it covered all the optimal possible schemes?
>
> We do not study all schemes in the paper to limit the number of curves in the paper. As said in the main notebook of the source code, *All details are given to extend our analysis to other numerical schemes.* We have added a sentence in the paper at line 80: *by straightforwardly generalizing our study to higher order linear numerical schemes.*
>
>
> > 2.For complex data distributions, analytic solutions may be difficult to obtain, which may affect the computational efficiency of the model. Does the paper consider the issue of computational efficiency in practical applications?
>
> Our paper can be applied to any Gaussian distribution, including one with a non-invertible covariance matrix. As discussed in the paper (*Section 6: Discussion and limitations*), we do not know how to generalize our results to compute exact Wasserstein error and derive exact solutions to the equations for more complex data distribution.

---

### Official Review · Reviewer_26fn · 2024-10-30

**Soundness:** 3
**Presentation:** 3
**Contribution:** 3
**Rating:** 8
**Confidence:** 3

**Summary:**

The paper presents a theoretical investigation of the different sources of errors in diffusion models in the case of Gaussian distributions. I.e., the setting is on purpose limited to this simple case in order to be able to derive closed form solutions to the forward and backwards diffusions and to provide a direct error investigation.

**Strengths:**

- the paper presents an interesting description and investigation of the sources of errors in diffusion models
- though restricted to specific cases, I found the discussions and results interesting. I believe the findings can be useful for practical application of diffusion models
- the paper is well written and nicely presented

**Weaknesses:**

- one can of course criticize the Gaussian assumption as being limited and not reflecting the complexity of real world data. I still believe the paper was an interesting read, even though it covers a specific case

**Questions:**

- I was a bit surprised that closed form solutions for the forward and backwards SDEs in the Gaussian case have not been presented in the literature earlier on. My guess would have been that they could be found in SDE textbooks. The authors provide some newer references to related results at the end of section 3. My question would be if you have thoroughly checked earlier SDE literature (before diffusion models became popular in the ML world) if these results have been covered previously?

---

> ### Author Response · Authors · 2024-11-17
>
> We thank  you for your positive reviews on our paper.
>
> > I was a bit surprised that closed form solutions for the forward and backwards SDEs in the Gaussian case have not been presented in the literature earlier on. My guess would have been that they could be found in SDE textbooks. The authors provide some newer references to related results at the end of section 3. My question would be if you have thoroughly checked earlier SDE literature (before diffusion models became popular in the ML world) if these results have been covered previously?
>
> To the best of our knowledge, we cannot find these precise results in the SDE literature. A main obstacle to this is the fact that most of the books present results in a 1D setting.

---

### Official Review · Reviewer_wE4j · 2024-11-02

**Soundness:** 3
**Presentation:** 3
**Contribution:** 1
**Rating:** 3
**Confidence:** 3

**Summary:**

Theory:
- It is well known that the W2-distance between two gaussian distributions is exactly computable. If we consider diffusion process described by a linear drift term, it is also well known that the corresponding integral kernel is also gaussian, which means if we consider **the initial distribution as gaussian (gaussian assumption)**, the remaining all marginal distributions at diffusion time t are also gaussian distributions as commented above of eq (10). The degrees of freedom of the distribution is described by covariance matrix $\Sigma_t$.
- The authors also provide solutions of backward SDE (Proposition 2) and backward probability flow ODE (Proposition 3) under the gaussian assumption, which provide exact expressions of each covariance matrix and covariance on $t=T$ with forward time expression.
- By considering normal distribution at $t=T$ with forward time expression, the above expressions provide covariance at time t with backward SDE and ODE, which provide an inequality of W2-distance (Proposition 4).
- In addition, the authors classify errors as:
    - the initialization error, which measures discrepancy between p_T and model's latent distribution,
    - the discretization error, which measures errors caused by numerical integral of SDE or ODE,
    - the truncation error, which measures an error caused by integral truncation at t=$\epsilon>0$.

Numerical experiments:
- They conducted some experiments to measure these errors in W2-sense (Table 2) and reported (each $\Sigma_t$'s) eigenvalue dependencies of each error with "gaussianized" CIFAR-10 data (Figure 2).
- They studied effects of score approximations by models (section 5).

**Strengths:**

- clarity(+)
    - The paper is well written and easy to follow. The theory part is easy to understand.
    - The numerical experiments are somewhat informative.

**Weaknesses:**

- significance (-?)
    - Thanks to the gaussian assumption, everything is very clear. However, at the same time, it makes the all results a little weak. I am not sure about how these exact results help in understanding the errors in actual learning outcomes. I summarized some questions about it below.

**Questions:**

I know that considering non-gaussian case is out of scope of this paper, but I would like to know the followings.
1. In the ablation study, the authors reported some influences based on the classified error categories (initialization/discretization/truncation errors), but it is not clear where the gaussian assumption is being used in the consideration of this error.
2. In Figure 2, as the authors commented, all errors around $\lambda=1$ drop suddenly. Why does it happen? In addition, similar sudden drops happen some other $\lambda$s. Are there any explanations on why it happens?
3. In the same page, the authors leave a comment "... for any Gaussian distribution Heun's method introduces nearly no additional discretization error, ..." Is this significantly influenced by the gaussian assumption, or can this be stated regardless of the gaussian distribution?
4. In page 9, to my understanding, the authors approximate W2 between $p_\theta$ and $p_{data}$ by replacing $p^{emp} \to p_\theta$ in eq (26). If so, how do you get $\lambda^{\xi, emp} = \lambda^{\xi, \theta}$ ? It would be related to Appendix F.3, but I'm not sure how to get $\Gamma$ in this case.
5. Besides these questions, if the authors have any insights indicating that the current results can be commonly applied outside the gaussian assumption, please make it clear.

---

> ### Author Response · Authors · 2024-11-17
>
> We thank you for your interesting questions on our paper. Nonetheless, you do not question the validity or the novelty of our work. Can you please explain more your negative rating (3: reject, not good enough) ?
>
> We answer your questions below.
>
>
> > 1. In the ablation study, the authors reported some influences based on the classified error categories (initialization/discretization/truncation errors), but it is not clear where the gaussian assumption is being used in the consideration of this error.
>
> The ablation study shows exact 2-Wassertein error compared to the exact backward solution. As discussed in section *Discussion and limitations* (more precisely, between lines  516 and 525), to the best of our knowledge, Gaussian assumption is necessary to compute exact Wasserstein errors and exact solutions to the equations.
>
> > 2. In Figure 2, as the authors commented, all errors around  drop suddenly. Why does it happen? In addition, similar sudden drops happen some other s. Are there any explanations on why it happens?
>
> For $\lambda = 1$, the diffusion model has nothing to do, indeed, the role of the diffusion model is to go from $\lambda_T = 1$ to $\lambda_0 = 1$ in the covariance matrix diagonalization basis. The other drops remain unexplained.
>
> > 3. In the same page, the authors leave a comment "... for any Gaussian distribution Heun's method introduces nearly no additional discretization error, ..." Is this significantly influenced by the gaussian assumption, or can this be stated regardless of the gaussian distribution?
>
> We work in the Gaussian case, where we can compute exact Wasserstein distances. Generalization to other cases remains to be investigated mathematically to confirm the experimental studies (as mentioned at line 77, *In particular, it confirms the strength of best practice scheme such as Heun's method for the ODE flow [Karras et al., 2022].*).
>
> > 4. In page 9, to my understanding, the authors approximate W2 between $p_{\theta}$ and $p_{\text{data}}$ by replacing $p^{\text{emp}} \rightarrow p_{\theta}$ in eq (26). If so, how do you get $\lambda^{\xi,\text{emp}} = \lambda^{\xi,\theta}$ ? It would be related to Appendix F.3, but I'm not sure how to get $\boldsymbol{\Gamma}$ in this case.
>
> Thank you for this interesting question.
>
> As mentioned at line 484: *Let us precise that this approximation underestimates the real Wasserstein distance since it wrongly assumes that the distributions $p_\theta^{\text{EM}}$, $p_\theta^{\text{Heun}}$ are Gaussian with a covariance matrix diagonalizable in the same basis than the covariance matrix $\Sigma$ of $ADSN(U)$.*
>
> To be more precise, by assuming that $p_\theta$ follows a Gaussian distribution $\mathcal{N}(0,\boldsymbol{\Gamma})$, given the size of the images, estimating $\boldsymbol{\Gamma}$ without an a priori on its structure is intractable. For this reason, we further assume that $\boldsymbol{\Gamma}$  admits the same eigenvectors as $\boldsymbol{\Sigma}$, that is the Fourier basis.
> Under this assumption, we evaluate the corresponding eigenvalues by using the reasoning of Appendix F.3.
>
>
>
> > 5. Besides these questions, if the authors have any insights indicating that the current results can be commonly applied outside the gaussian assumption, please make it clear.
>
> Again, we work on the Gaussian setting where we are able to demonstrate new results.

---

> > ### Comment · Reviewer_wE4j · 2024-11-27
> >
> > I would like to thank the authors to respond my concerns.
> > Let me leave some comments to your replies:
> >
> > - on Gaussian stuffs
> >
> > As far as I understand, the authors did not answer anything beyond Gaussian assumption. I appreciate the authors' honestness, and I understand the scope of this paper is no more than it. However, I think that the readers/attendees expect something informative on real world machine learning problem even if the presentation focuses on some theoretical stuffs. Could the authors provide something in this sense?
> >
> > - on question 4 in my review
> >
> > Thank you for clarifying the situation. Now I think I understand what the authors did.
> >
> > ---
> >
> > Next, let me answer to the following comment in the authors rebuttal:
> >
> > > Nonetheless, you do not question the validity or the novelty of our work. Can you please explain more your negative rating (3: reject, not good enough) ?
> >
> > On the validity or the novelty, my evaluations are in average, so I could not count them as neither positive nor negative evaluations.
> > My negative rating mainly comes from the low significance caused by the Gaussian assumption. I I think the Gaussian assumption itself is OK, but want to know something related to actual learning outcomes as I asked in the above part of this reply.

---

> > > ### Author Response · Authors · 2024-11-28
> > >
> > > Thank you for your response.
> > >
> > > The ambition and novelty of our article lie in the study of the exact errors of diffusion models for a family of distributions. This motivated us to adopt the strong Gaussian assumption. In this framework, we have access to exact error plots and precise insights into the influence of different error types.
> > >
> > > We wish to emphasize that bridging the gap between theoretical studies and practical implementations remains a significant challenge. The state-of-the-art research addressing the convergence of diffusion models typically involves inequalities between distributions that result in bounds that are difficult to interpret in practice. Moreover, such studies often introduce assumptions that are generally impossible to verify empirically. In our work, we opted for a stronger, practically verifiable assumption, which allowed us to trace exact convergence errors. Nevertheless, as is the case in the literature, a substantial gap remains between theoretical analysis and the concrete practical applications of diffusion models.

---

### Official Review · Reviewer_BPJ2 · 2024-11-04

**Soundness:** 2
**Presentation:** 2
**Contribution:** 2
**Rating:** 3
**Confidence:** 4

**Summary:**

The authors define a centered Gaussian $N(0, \Sigma)$ as the data distribution and assume that the inference SDE, or forward process, is VP-SDE. VP-SDE is a linear diffusion process and it’s transition kernel, a Gaussian distribution, can be computed in closed-form. This allows the authors to

1. Compute the exact score $\nabla \log q(x_t)$ since the process $x_t$ for all time $t$ is Gaussian
2. Having access to a closed-form for the score allows one to compute exact solutions to the backward SDE (prop 2) and the probability flow ODE (prop 3), which then allows one to analyze the effect of:
    1. impact of initialization error in the ODE and SDE generative models
    2. impact of discretization error
    3. impact of truncation error
    4. influence of the eigenvalues of the covariance matrix

Using the exact derivations, the authors then analyze a Gaussian data distribution with the covariance computed from samples from the CIFAR10 dataset. The authors analyze the $W_2$ error of different samplers on this dataset and present results in table 2.

In their next experiment, the authors learn the score function using a U-Net and see if their findings translate to when the score function architecture is nonlinear in a setting where the exact score is available as well.

**Strengths:**

In section 4, particular table 2, the authors provide a thorough analysis of the impact of the:

1. initialization error
2. truncation error
3. discretization error

for commonly used samplers such as EM, Heun and Euler in the centered Gaussian case.

**Weaknesses:**

While the authors have developed a simple framework for analyzing the different sources of error while sampling, it is limited to a Gaussian data distribution. It is not made clear how these insights generalize beyond Gaussian data and fits into the wider literature. See questions.

The score function is available in closed-form when the data distribution is a mixture of Gaussians, and as the authors mention the Wasserstein error is not computable in closed form. However, empirical metrics/divergences such as MMD, kernel Stein discrepancies, etc are available.

Citations:

1. Proposition 1 has also been derived in appendix A in [Albergo 2023], where the authors derive the score for a number of different forward diffusion processes, flows, etc. Can the authors cite them in the main paper?
2. Similarly, for proposition 3, a solution to the PF-ODE has been derived in appendix B in [Khrulkov 2022]. Can the authors cite them in the main paper?

**Questions:**

1. Can the authors add appropriate citations for proposition 1, for instance eq 6.19 and 6.20 in [SARKKA & SOLIN 2019].
2. The ADSN data samples are out of distribution for the classifier used to compute FID scores. Could the authors use a different metric? for instance MMD or other metrics/divergences.

As mentioned in the weaknesses section, the findings of this paper are not clearly situated within the wider literature, that is what this paper is adding beyond what is known already. For instance,

1. Error due to discretization is a widely studied problem in numerical sampling. Heun’s is a second-order integration scheme leading to slower accumulation of error compared to Euler’s method. Would the Gaussian data analysis agree with the fact that  a higher order scheme such as RK4 provide a more accurate integration of the PF-ODE?
2. In line 502-503, the authors mention that “the score approximation error is by far the most impactful source of error”. However, it is already known that score estimation error can lead to an error in sampling, for instance in [Chen et al 2023], [De Bortoli et al 2021], etc.
3. Can the Gaussian data analysi provide any theoretical justification or insight into the robustness of SDE sampling over ODE sampling besides empirical evidence?

**References**

[Chen et al 2023] Chen, Sitan, Sinho Chewi, Jerry Li, Yuanzhi Li, Adil Salim, and Anru R. Zhang. "Sampling is as easy as learning the score: theory for diffusion models with minimal data assumptions." *arXiv preprint arXiv:2209.11215* (2022).

[De Bortoli et al 2021] De Bortoli, V., Thornton, J., Heng, J. and Doucet, A., 2021. Diffusion schrödinger bridge with applications to score-based generative modeling. Advances in Neural Information Processing Systems, 34, pp.17695-17709.

---

> ### Author Response · Authors · 2024-11-17
>
> We thank you for your comments on our paper. Most of your remarks concern minor bibliographical complements that improve the paper and that we are happy to include.
>
> Can you please explain more your negative rating (3: reject, not good enough) ?
>
> We answer your questions below.
>
> > Proposition 1 has also been derived in appendix A in [Albergo 2023], where the authors derive the score for a number of different forward diffusion processes, flows, etc. Can the authors cite them in the main paper?
>
> Thank you for the reference to the preprint "Stochastic Interpolants: A Unifying Framework for Flows and Diffusions", Albergo, Boffi, Vanden-Eijnden,  2023. Indeed, the score is derived in Appendix A (Equation A.8) in the Gaussian case and we have added the citation at line 238.
>
>
> > Similarly, for proposition 3, a solution to the PF-ODE has been derived in appendix B in [Khrulkov 2022]. Can the authors cite them in the main paper?
>
> We believe we cite the published version of this paper (Khrulkov et al., ICLR 2023) at line 247.
>  : *Indeed,  (Khrulkov et al., 2023) derive the solution of the flow ODE under Gaussian assumption at infinite time horizon  (Khrulkov et al., 2023, Appendix B).*
>
> > Can the authors add appropriate citations for proposition 1, for instance eq 6.19 and 6.20 in [SARKKA & SOLIN 2019].
>
> We already cite [SARKKA & SOLIN 2019] at line 237. We can indeed add the more precise reference to equation number, as done in the new version.
>
> > The ADSN data samples are out of distribution for the classifier used to compute FID scores. Could the authors use a different metric? for instance MMD or other metrics/divergences.
>
> We have made the choice to use the standard metric. All the reported FID values vary consistently with other Wasserstein metrics.
>
>
> >  Would the Gaussian data analysis agree with the fact that a higher order scheme such as RK4 provide a more accurate integration of the PF-ODE?
>
> We do not study RK4 to limit the number of curves in the paper. As said in the main notebook of the source code, *All details are given to extend our analysis to other numerical schemes.* We have added a sentence in the paper at line 80: *by straightforwardly generalizing our study to higher order linear numerical schemes.*
>
> > In line 502-503, the authors mention that “the score approximation error is by far the most impactful source of error”. However, it is already known that score estimation error can lead to an error in sampling, for instance in [Chen et al 2023], [De Bortoli et al 2021], etc.
>
>  We already cite these papers in the introduction but we can appropriately cite them in the score approximation section (see line 503 in the new version). Let us stress on the fact that we provide a more precise reasoning under Gaussian assumption with exact Wasserstein errors of the other error types.
>
>
> > Can the Gaussian data analysi provide any theoretical justification or insight into the robustness of SDE sampling over ODE sampling besides empirical evidence?
>
>
> As written at line 505: *We may explain this behavior by recalling the results of Proposition 4 that shows that SDE solutions are less sensitive to initialization errors than ODE.
> Indeed, adding noise at each iteration tends to mitigate the accumulated errors, and score approximation may be considered as some initialization error occurring at each step.*

---

> > ### Comment · Reviewer_BPJ2 · 2024-11-17
> >
> > Thank you for your response and adding the appropriate citations. I am willing to increase my score if the authors can address the central weakness highlighted in the first review: the findings of the paper are not clearly situated within the wider literature on both diffusion-based generative modeling and numerical methods, which I elaborate on below.
> >
> >
> > Can the authors situate the theoretical and empirical insights they provide about diffusion-based generative modeling ***within*** the existing literature. That is:
> >
> > - Can the authors explain what ***additional insights*** are provided by considering a Gaussian data distribution ***beyond what is already known from prior works***?
> >
> > For instance, here are some commonly known facts in the diffusion and numerical methods literature:
> >
> > - Higher-order sampling schemes, such as Heun’s and RK4, integrate more accurately than lower-order sampling schemes such as Euler’s method. This was demonstrated in several works on high-dimensional distributions [Zhang et al 2023, Dockhorn et al 2022] and as the authors themselves cite [Karras et al 2022].
> > - The effect of time-steps on the sampling scheme is also a widely studied topic in numerical integration. The authors own experiments agree with the well-known fact from numerical analysis that a higher-order scheme with small step-size will incur less error. Is the Gaussian data framework adding any additional insight beyond what is established in the numerical integration literature and diffusion model sampling literature.
> > - Score approximation error is important since the learned score determines the probability distribution the model samples. Prior work [Esser et al 2024] shows that the denoising score matching loss, computed on a validation set, correlates with sample quality, prompt alignment, etc. showing that the score approximation error is vital for sample quality.
> >
> > What additional insight is gained from considering Gaussian data distributions beyond these insights which are already known in the literature.
> >
> > In the lines below, do the authors mean Heun’s is the best sampler out of the 4 samplers they consider or in general. Secondly, would it be true that a higher-order scheme such as RK4 is better than Heun’s method?
> >
> > 1. Lines 535-536: This theoretical analysis led to conclude that Heun’s scheme is the best numerical solution, in accordance with empirical previous work (Karras et al., 2022).
> > 2. Line 23-25: our experiments show that the recommended numerical schemes from
> > the diffusion models literature are also the best sampling schemes for Gaussian
> > distributions.
> >
> > [Zhang et al 2023] Improved Order Analysis and Design of Exponential Integrator for Diffusion Models Sampling
> >
> > [Esser et al 2024] Scaling rectified flow transformers for high-resolution image synthesis
> >
> > [Dockhorn et al 2022] Dockhorn, Tim, Arash Vahdat, and Karsten Kreis. "Genie: Higher-order denoising diffusion solvers." *Advances in Neural Information Processing Systems* 35 (2022): 30150-30166.
> >
> > [Karras et al 2022] Karras, Tero, et al. "Elucidating the design space of diffusion-based generative models." *Advances in neural information processing systems* 35 (2022): 26565-26577.

---

> > > ### Author Response · Authors · 2024-11-19
> > >
> > > At the risk of being repetitive with the introduction of our paper, to the best of our understanding, all the mentioned references provided insights using either theoretical approaches with asymptotic or non-asymptotic upper bounds, which are far from being tight quantifications, or by using experimental validation on a few datasets and mostly relying on feature distance evaluation like FID.
> > >
> > > The bounds provided in the reference you mentioned [Zhang et al 2023] primarily concern numerical schemes. In [Zhang et al 2023, Esser et al 2024, Dockhorn et al 2022, Karras et al 2022], the distance between the data distribution and the sampled distribution is empirical, considering FID. Our code provides exact Wasserstein errors rather than bounds, allowing for a more precise evaluation of the schemes. We provide exact 2-Wasserstein distances in data space for all Gaussian distributions, along with a precise quantification for the various sources of errors.
> > >
> > > Moreover, despite strong empirical results, the approaches of previous studies [Zhang et al 2023, Esser et al 2024, Dockhorn et al 2022, Karras et al 2022] involve lengthy sampling times and expensive tests to evaluate the efficiency of a numerical scheme. In contrast, our test on Gaussian distributions is computationally inexpensive and can serve as a foundation for testing other schemes in the future. Our code can be adapted to test other time schedules or schemes. We envision that someone tests their schemes with our code before sampling a large dataset of images. In summary, our paper offers a fast and flexible testing framework for numerical schemes across an entire family of distributions.
> > >
> > > About RK4, we changed the sentence you cite by adding 'out of the four considered schemes' (see line 535). Again, it is possible to adapt the code to include the evaluation of RK4. We chose not to consider this scheme to avoid a large number of NFE. Furthermore, Heun's scheme already achieves a highly satisfactory error, given its simplicity of implementation (see Figure 1.(a)).
> > >
> > > In addition, we believe that the conclusion of Proposition 4 is new and the clarifications of Proposition 3 are insightful.
> > >
> > > The fact that we did not find behavior contradictory to what is accepted as common knowledge from prior work is reassuring and should not be interpreted as a weakness of our contribution.

---

> > > > ### Comment · Reviewer_BPJ2 · 2024-11-19
> > > >
> > > > My primary concern I believe is still is not addressed by the authors:
> > > >
> > > > - Can the authors explain what ***additional insights*** are provided by considering a Gaussian data distribution ***beyond what is already known from prior works***?
> > > >
> > > > > “About RK4, we changed the sentence you cite by adding 'out of the four considered schemes' (see line 535). Again, it is possible to adapt the code to include the evaluation of RK4. “
> > > > >
> > > >
> > > > My question regarding RK4 was not about conducting an additional experiment. It was to highlight the fact that discretization errors in numerical integration schemes has been an active area of research and can provide guidance on integration schemes, such as
> > > >
> > > > 1. Showing that RK4 is in general a better integration scheme than Euler’s or Heun’s methods.  This has been shown in countless experiments.
> > > > 2. Smaller time steps are in general better for any integration scheme.
> > > >
> > > > **Q: What does using the proposed Gaussian data framework add over these two facts?**
> > > >
> > > > > “We envision that someone tests their schemes with our code before sampling a large dataset of images. In summary, our paper offers a fast and flexible testing framework for numerical schemes across an entire family of distributions.”
> > > > >
> > > >
> > > > Testing on toy, or simpler, distributions has been a practice in generative modeling research since it has existed. Several diffusion modeling papers already experiment with toy distributions, which are inexpensive and have some complexities of real data distributions, such as disjoint supports, etc. For instance, see
> > > >
> > > > 1. [Lipman et al 2023, Singhal et al 2024, Albergo et al 2022] Experiment with the checkerboard dataset
> > > > 2. [Albergo et al 2024] derive the exact scores for transporting a mixture of Gaussian to another mixture of Gaussian for any interpolants, including all linear diffusion processes.
> > > > 3. Prior works such as [Singhal et al 2024] conduct low-dimensional experiments where they report MMD metrics to compare choices in training diffusion models. MMD is an integral probability metric which can measure convergence in distribution.
> > > >
> > > > **Q: What is the proposed Gaussian data framework a better framework?**
> > > >
> > > > > “The fact that we did not find behavior contradictory to what is accepted as common knowledge from prior work is reassuring and should not be interpreted as a weakness of our contribution.”
> > > > >
> > > >
> > > > My primary question in the review: is what *new* fact, method, framework, etc this paper proposes over what is already known and *why* is that better than what exists.
> > > >
> > > > [Lipman et al 2023] Flow matching for generative modeling
> > > >
> > > > [Singhal et al 2024] What's the score? Automated Denoising Score Matching for Nonlinear Diffusions
> > > >
> > > > [Albergo et al 2022] Building Normalizing Flows with Stochastic Interpolants
> > > >
> > > > [Albergo et al 2024] Stochastic Interpolants: A Unifying Framework for Flows and Diffusions

---

> > > > > ### Author Response · Authors · 2024-11-22
> > > > >
> > > > > We are sorry that you still have concern regarding the interest and novelty of our work.
> > > > >
> > > > > We establish several new mathematical results that we believe provide new insight and new generic tools to the community by fully treating the example of Gaussian distributions.
> > > > >
> > > > > The novelty of our work is summarized between lines 70 and 75:
> > > > > - We give the exact solutions for both the backward SDE and the probability flow ODE
> > > > > - We fully describe the Gaussian processes that occur when using classical sampling discretization schemes.
> > > > > - We derive exact 2-Wasserstein errors for the corresponding sample distributions and are able to assert for the influence of each error type on these errors, as illustrated by Figure 1.
> > > > >
> > > > > The score function has already been computed exactly for Gaussian mixtures [Albergo et al., 2024] and evaluations of convergence in probability are proposed in prior works [Lipman et al., 2023; Singhal et al., 2024; Albergo et al., 2022; Albergo et al., 2024], relying on metrics empirically computed at the endpoint of the backward process.
> > > > >
> > > > > The Gaussian assumption provides the following contributions:
> > > > > - It enables the determination of the exact solution for the backward process.
> > > > > - It allows for the exact computation of the theoretical Wasserstein distance at any time during the backward process.
> > > > > - It provides an exact observation of the influence of each type of error on the convergence of the diffusion model.
> > > > >
> > > > > Unlike previous works, the error is analyzed continuously over time and does not rely on empirical estimations, which appears to be a different approach.

---

> > > > > > ### Comment · Reviewer_BPJ2 · 2024-11-23
> > > > > >
> > > > > > > The score function has already been computed exactly for Gaussian mixtures [Albergo et al., 2024] and evaluations of convergence in probability are proposed in prior works [Lipman et al., 2023; Singhal et al., 2024; Albergo et al., 2022; Albergo et al., 2024], relying on metrics empirically computed at the endpoint of the backward process.
> > > > > >
> > > > > > [Singhal et al., 2024] compute MMD not just at the endpoints but during the entire sampling process. MMD is an integral probability metric and for low-dimensional experiments, it is easy to get low variance estimates. I also note that [Singhal et al., 2024] provide the mean and covariance of the transition kernel of affine SDEs, which is what the backward SDE is when the score is linear.
> > > > > >
> > > > > > > We establish several new mathematical results that we believe provide new insight and new generic tools to the community by fully treating the example of Gaussian distributions.
> > > > > > Can the authors list these new insights as a reply here?
> > > > > >
> > > > > >
> > > > > > While I do not disagree that the authors might be the first to write the exact solutions of the PF-ODE, which is a linear affine ODE from my understanding, and the Gaussian processes implied by numerical samplers, the authors have not taken any efforts to explain what insights their paper provides over existing approaches that are common in the field already.
> > > > > >
> > > > > > The sources of error considered are already well studied and the insight gained by the authors is that a higher-order scheme with a small step size is the sampler which accumulates the least error. However, this has been well known in the field of numerical methods.
> > > > > >
> > > > > > I am willing to raise my score if the authors can provide an answer to my questions.

---

> > > > > > > ### Author Response · Authors · 2024-11-25
> > > > > > >
> > > > > > > We honestly attempted to respond several times to your concerns.
> > > > > > > We are sorry that you do not find our answers satisfying and that you stand by your negative evaluation due to a lack of "additional insight".
> > > > > > > We do believe that our new mathematical results are of interest for the various reasons enumerated above.

---

### Author Response · Authors · 2024-11-17
**Global rebuttal**

We thank all the reviewers for their comments on our paper. Despite mixed reviews, none of the reviewers question the validity of the demonstrated results. As discussed in the paper, the Gaussian assumption framework indeed implies a practical limited scope for our results. Nonetheless, we would like to emphasize the new theoretical contributions to the study of diffusion models, such as Proposition 2, 3 and 4. To the best of our knowledge, it is novel to obtain exact solution to the backward SDE of a diffusion model and provide strong theoretical results comparing the sampling of the backward SDE and the probability flow ODE.

We believe that our work fits into the literature by balancing the practical and empirical study of diffusion models and bringing novelty in the convergence study of diffusion models by computing the **exact** Wasserstein distances.
Our work is of interest for both experimental and theoretical researchers in the field.
For practioners, it provides a full parametric family of distributions with known exact solutions that can serve as calibration for new numerical schemes. For theoretical studies, the same solutions and associated exact Wasserstein errors provide insights on the tightness of the upper bounds established in convergence theorems for diffusion models.

---

### Meta-Review · Area_Chair_VLRT · 2024-12-12

**Metareview:**

The paper provided exact solutions to the diffusion model under the Gaussian distribution, and the associated error in Wasserstein distance due to the tractable form of Gaussian. However, the main drawback of the paper is that diffusion for the Gaussian distribution is the least interesting distribution from both theoretical and practical perspectives, since the main fight ground for diffusion models is for multi-modal distributions. Therefore, this paper has limited novelty and is of limited interest to the ICLR readership.

**Additional Comments On Reviewer Discussion:**

The most heated discussion is around the significance and validity of the Gaussian assumption, and if this work brings new insights to the practice of diffusion models.

---

### Decision · Program_Chairs · 2025-01-22

Reject